



# Below-cloud scavenging of aerosol by rain: A review of numerical modelling approaches and sensitivity simulations with mineral dust

Anthony C. Jones[1], Adrian Hill[1], John Hemmings[1], Pascal Lemaitre[2], Arnaud Quérel[3], Claire L. Ryder[4], and Stephanie Woodward[1]

[1] Met Office, Fitzroy Road, Exeter, EX1 3PB, UK
[2] Institute for Radiation Protection and Nuclear Safety (IRSN), PSN-RES, SCA, LPMA, Fontenay-aux-Roses, 92260, France
[3] Institute for Radiation Protection and Nuclear Safety (IRSN), PSE-SANTE, SESUC, BMCA, Fontenay-aux-Roses, 92260, France
[4] Department of Meteorology, University of Reading, RG6 6BB, Reading UK

*Correspondence to*: Anthony C. Jones (anthony.jones@metoffice.gov.uk)

**Abstract.** Theoretical models of the below-cloud scavenging (BCS) of aerosol by rain yield scavenging rates that are 1-2 orders of magnitude smaller than observations and associated empirical schemes for submicron-sized aerosol. Even when augmented with processes which may explain this disparity, such as phoresis and rear-capture in the raindrop wake, the theoretical BCS rates remain an order of magnitude less than observations. Despite this disparity, both theoretical and empirical BCS schemes remain in wide use within numerical aerosol models. BCS is an important sink for atmospheric aerosol, in particular for insoluble aerosol such as mineral dust which is less likely to be scavenged by in-cloud processes than purely soluble aerosol. In this paper, various widely used theoretical and empirical BCS models are detailed and then applied to mineral dust in climate simulations with the Met Office's Unified Model in order the gauge the sensitivity of aerosol removal to the choice of BCS scheme. We show that the simulated accumulation mode dust lifetime ranges from 5.4 days in using an empirical BCS scheme based on observations to 43.8 days using a theoretical scheme while the coarse mode dust lifetime ranges from 0.9 to 4 days, which highlights the high sensitivity of dust concentrations to BCS scheme. We also show that neglecting the processes of rear-capture and phoresis may overestimate submicron-sized dust burdens by 83 %, while accounting for modal widths and mode-merging in modal aerosol models alongside BCS is important for accurately reproducing observed aerosol size distributions and burdens. This study provides a new parameterisation for the rear-capture of aerosol by rain and is the first to explicitly incorporate the rear-capture mechanism in climate model simulations. Additionally, we answer many outstanding questions pertaining to the numerical modelling of BCS of aerosol by rain and provide a computationally inexpensive BCS algorithm that can be readily incorporated in other aerosol models.

## 1 Introduction

Atmospheric aerosols play an important role in climate system by altering energy fluxes, interacting with clouds, transferring nutrients to ecosystems, and contributing to atmospheric chemistry and air quality (Haywood and Boucher, 2000). For these reasons, it is vital that aerosol microphysical processes are accurately modelled in General Circulation Models (GCMs),





especially given that aerosol-climate interactions are one of the leading causes of uncertainty in existing GCMs (Carslaw *et al.*, 2013). Aerosols are efficiently removed from the troposphere by wet deposition processes such as in-cloud scavenging

(ICS) (also denoted 'rainout' or 'nucleation scavenging') and below-cloud scavenging (BCS) (also denoted 'washout' or 'impaction scavenging') (Pruppacher and Klett, 2010). ICS occurs when aerosols act as cloud condensation nuclei and form cloud droplets or ice crystals which then grow and fall as precipitation, or when aerosols collide with existing cloud droplets. BCS occurs when falling hydrometeors, such as rain or snow, irreversibly collect ambient aerosol in their path. The BCS rate strongly depends on the rain intensity, rain droplet size distribution, and the collection efficiency between raindrop and aerosol

particle (Laakso *et al.*, 2003).

A long-established problem in BCS modelling is reconciling scavenging rates from *in situ* atmospheric observations with rates derived from conceptual models and laboratory experiments (Beard, 1974; Davenport and Peters, 1978; Radke *et al.*, 1980; Volken and Schumann, 1993; Laakso *et al.*, 2003). In particular, BCS rates from theoretical models are 1-2 orders of magnitude

smaller than observed rates for accumulation sized (diameters of $0.1 \leq d_p \leq 1$ µm) particles (Wang *et al.*, 2010). Given that accumulation aerosols are particularly important to the climate system for cloud microphysics, radiative interactions, heterogeneous chemistry, air quality, and myriad other climate interactions, it is important to represent aerosol microphysics accurately in GCMs. The accumulation size range, where BCS scavenging rates exhibit a global minimum owing to the lack of a dominant scavenging process, is widely denoted the "Greenfield gap", and the scavenging minimum is seen in both

observations and theory, albeit with different magnitudes (Greenfield, 1957).

Various hypotheses have been put forward to explain the disparity between observations and theory. Beard (1974) and Davenport and Peters (1978) suggested that aerosol hygroscopic growth and electrostatic charge effects, which are not explicitly modelled by the early theoretical models, may explain the disparity. Quérel *et al.* (2014) highlighted the effect of

downdrafts caused by the falling precipitation on near-surface aerosol concentrations, with comparatively clean air transported downward from aloft possibly masking the direct BCS effect. Additional uncertainty arises from modelling BCS by millimetre sized raindrops given their tendency to oscillate in freefall, with complex flows leading to enhanced rear-capture and frontal-capture effects (Wang and Pruppacher, 1977; Lemaitre *et al.*, 2017). Although atmospheric turbulence has been imputed for the disparity between observations and theory (e.g., Wang *et al.*, 2010, 2011), Vohl *et al.* (2001) found little impact of

turbulence on BCS rates in their laboratory experiments with larger rain droplets (diameters $D_d \geq 600$ µm). A recent hypothesis is that the enhanced BCS rates from observations may be due to contributions from ICS and other confounding atmospheric processes such as turbulent diffusion, given that it is difficult to conduct a controlled BCS experiment in the actual atmosphere (Andronache *et al.*, 2006; Wang *et al.*, 2011). Indeed, BCS rates determined from the controlled 'outdoor' experiment of Sparmacher *et al.* (1993), in which monodisperse aerosol in a wind-shielding chamber was subjected to natural

precipitation, were much closer to theoretical values than other observational values (Wang *et al.*, 2010).





The disparity between observed and theoretical BCS rates has stimulated a wide range of approaches of varying complexity for modelling BCS in GCMs (Jung *et al.*, 2003; Croft *et al.*, 2009, 2010; Wang *et al.*, 2010, 2014). The most widely utilised theoretical BCS approach in GCMs is to follow Slinn (1984) in expressing the raindrop-particle collection efficiency – an important BCS parameter representing the ratio of number of collisions between a rain droplet and particles to the total number of particles in an area equal to the raindrop's cross-sectional area – as a linear combination of collection efficiencies due to Brownian motion, inertial impaction, and interception (Seinfeld and Pandis, 1998; Jung *et al.*, 2003; Loosmore *et al*., 2004; Berthet *et al.*, 2010; Wang *et al.*, 2010). Slinn (1984) proposed formulae for the individual collection efficiencies based on data from laboratory experiments and dimensional analyses. Other processes are known to contribute to BCS including thermophoresis and diffusiophoresis, by which particles move along temperature and water gradients respectively, and attraction between oppositely charged raindrops and particles (Slinn and Hales, 1971; Davenport and Peters, 1978; Andronache *et al*., 2004, 2006). Recently, Lemaitre *et al.* (2017) compared results from historical numerical models, with and without the assumption of Stokes flow, to derive an empirical formula for the collection efficiency in the recirculating flow of the raindrop's wake. Lemaitre *et al.* (2017) and Quérel *et al.* (2014) have proposed that this 'rear-capture' effect, neglected by Slinn (1984), be directly added to the established processes in BCS schemes. Wang *et al*. (2010) recommended that the theoretical schemes which yield the highest BCS rates be used in GCMs, while Wang *et al.* (2014) develop on this suggestion by deriving a semi-empirical formula for the 90% percentile of theoretical BCS rates from the literature.

An alternative approach to the theoretical modelling of Slinn (1984) and others (e.g., Hall, 1980; Flossmann, 1986) for deriving BCS rates ($\Lambda$) is to empirically fit formulae to observations. Laakso *et al.* (2003) measured $\Lambda$ over 6 years at a boreal forest site in Southern Finland, and then combined these measurements with similar observations from Volken and Schumann (1993), to derive a widely utilised empirical fit for $\Lambda$ as a function of aerosol size and rain intensity. A similar approach was conducted by Baklanov and Sørensen (2001) who omitted a size dependence in the formulation of $\Lambda$ for Aitken (diameters of $0.01 \leq d_p \leq 0.1$ μm) and accumulation sized aerosols. Therein lies the issue with empirical schemes; notably, what to do outside the boundaries of observations. Additionally, rain types differ with location (e.g., in terms of the electric charge density of raindrops) and aerosols differ in composition, and so the general applicability of empirical schemes fit to data in a single location is questionable (Wang *et al.*, 2014). Note though, that similar uncertainties are also present in the theoretical models, which are fit to laboratory data and observations (e.g., the raindrop number distribution is often parameterised as a function of rainfall rate, which is often fit to observations). Size-resolved $\Lambda$ from field data have become increasingly available in recent decades (e.g., Maria and Russell, 2005; Zikova and Zdimal, 2016; Blanco-Alegre *et al.*, 2018, 2021; Cugerone *et al.*, 2018; Lu *et al.*, 2019; Xu *et al.*, 2019) and are generally commensurate between campaigns across the aerosol size spectrum.

The panoply of BCS models used by the aerosol modelling community raises the question of what the implications are of selecting certain BCS models over others. Indeed, it would be useful for the aerosol modelling community to have the following Key Questions (KQ) answered before designing or selecting a BCS scheme:



KQ1.    To what extent does the use of an empirical BCS model over a theoretical model change atmospheric aerosol concentrations in a GCM?

KQ2.    How important is it to include missing processes in the Slinn (1984) BCS model, notably phoresis and the rear-capture effect? The rear-capture model of Lemaitre *et al.* (2017) is only valid for a narrow range of aerosol diameters, and thus an improved model - valid for the entire aerosol size spectrum – will be provided and utilised in this study.

KQ3.    Pertaining to modal aerosol schemes, to what extent does the use of a single moment BCS approach - where $\Lambda$ is computed solely using the aerosol modal median diameter while the width of the mode is ignored - over a double moment approach change simulated aerosol concentrations?

KQ4.    Pertaining to double-moment modal aerosol schemes, how important is it to include downward mode merging – or the redistribution of aerosol mass and number from a large to a neighbouring smaller mode – alongside BCS?

KQ4 requires further explanation. Many GCMs participating in the AeroCom phase III model intercomparison project employ a double-moment modal aerosol scheme (Gliß *et al.*, 2021). Modal schemes have the advantage over bulk schemes that the aerosol size distribution is permitted to evolve, albeit often within a predefined size bracket and – in the case of double-moment schemes – assuming a fixed modal width. Atmospheric processes such as coagulation, condensation, BCS, ICS and sedimentation may cause neighbouring modes to evolve such that they overlap and become indistinguishable (Whitby, 2002). Additionally, size-dependent processes such as BCS may alter the width of the ambient size mode, and thus a double moment modal aerosol scheme with fixed geometric widths would be unable to capture this effect. To account for this deficiency in the double-moment architecture, "mode merging" schemes are often employed to redistribute aerosol mass and number between neighbouring modes (Mann *et al.*, 2010). Given the highly size dependent nature of BCS, it is useful to test the impact of representing *downward mode merging* (i.e., the transfer of mass and number from the coarse mode to the smaller accumulation mode when the modes overlap) to account for contraction of the coarse mode as a result of BCS.

To answer the KQs, 20-year integrations are performed with the Met Office's Unified Model (UM) in a climate configuration, where the sole variable is the formulation of BCS applied to mineral dust aerosol. The UM represents aerosol using the double-moment Global Model of Aerosol Processes (GLOMAP-mode) model, which is coupled to the United Kingdom Chemistry and Aerosol (UKCA) model in the UM and cumulatively denoted UKCA-mode (Mann *et al.*, 2010; Mulcahy *et al.*, 2018; Jones *et al.*, 2021). Whilst UKCA-mode has in-built functionality to represent mineral dust in 2 insoluble modes representing accumulation and coarse (diameters of $d_p \geq 1$ µm) sized particles, this scheme has never been the default option in the Met Office Global Atmosphere science configuration - which forms the physical atmosphere in the UK's Earth System model - owing to lack of fidelity between simulations and observations, with UKCA-mode dust exhibiting too high dust concentrations away from source regions. Inefficient wet removal is thought to be an important factor, which may in part be addressed by the results of this study. Instead, the 6-bin dust scheme within the single-moment Coupled Large-scale Aerosol Simulator for



Studies in Climate (CLASSIC) aerosol framework (Woodward, 2001), remains the default option in Global Atmosphere version 7.1 (GA7.1) and later versions (Mulcahy *et al.*, 2020). UKCA-mode dust thus appears an ideal candidate for comparing BCS schemes, given its significant potential for improvement. However, the focus of this paper is to look at the underlying BCS theory using the UM, rather than to provide a direct comparison with existing functionality in this model.

The aim of this study is to outline the most widely utilised numerical BCS models and then to compare them quantitatively in UM simulations. In Section 2, various BCS models are presented and a computationally inexpensive BCS algorithm is proposed that can be readily incorporated in other GCMs. In Section 3, the box model simulations are described, while in Section 4 the UM configuration is described and the UM simulations are outlined. In Section 5, the various numerical BCS approaches are compared using the results of offline box model and UM simulations, in terms of spatiotemporal dust

concentrations and deposition rates. In Section 6, the results and implications of this study are discussed.

## 2 Below-cloud scavenging approaches

### 2.1 Overview of a new BCS algorithm

Fully resolved BCS schemes are computationally expensive to run in GCMs owing to the need to integrate BCS rates over the aerosol and rain droplet size distributions at every timestep and in every grid-cell that is subject to precipitation. Methods to

explicitly compute BCS online include the use of Gauss quadrature (e.g., Berthet *et al.*, 2010) or by simplifying the BCS equation to a polynomial in the aerosol diameter ($d_p$) and then using the moment method to obtain an analytical solution (e.g., Jung *et al.*, 2003). Alternatively, to reduce the computational cost, the BCS coefficient can be calculated offline as a function of aerosol and rain droplet size properties and standard atmospheric conditions, and then tabulated for simple interpolation in a GCM, which is the approach adopted here.


A new BCS algorithm, which has the quality that it is easy to change the underlying BCS parameterisation, is presented in this section. The time-dependent removal of aerosol by BCS is generally expressed as a first order decay equation (Seinfeld and Pandis, 1998; Wang *et al.*, 2010).

$$\frac{dn(d_p)}{dt} = -\Lambda(d_p)n(d_p) \tag{Eq. 1}$$

$$\Lambda(d_p) = \int_0^\infty \frac{\pi}{4} D_d{}^2 U_t(D_d) E(d_p, D_d) N(D_d) dD_d \tag{Eq. 2}$$

In Eq. 1, $n(d_p)$ is the size-resolved particle number concentration at time $t$, $d_p$ is the particle diameter, and $\Lambda(d_p)$ is the size-resolved scavenging coefficient or 'BCS rate'. Equation 2 expresses $\Lambda(d_p)$ as the integral of the collection kernel $K(d_p, D_d) =$



$\frac{\pi}{4}D_d{}^2 U_t(D_d)E(d_p, D_d)$ over the rain droplet size distribution, where $E(d_p, D_d)$ denotes the particle collection efficiency and $U_t(D_d)$ denotes the raindrop's fall speed. Generally, it is empirically assumed that the collection efficiency equals the collision efficiency or that all collisions between a hydrometeor and a particle result in successful collection (Weber *et al.*, 1969).

For the BCS scheme based on Slinn's (1984) model for $E(d_p, D_d)$ (Sections 2.2-2.4), $U_t(D_d)$ is parameterised following the
'gold standard' method of Beard (1976) (see Section S1 in the Supplement). In short, $U_t(D_d)$ is determined for 3 different rain droplet regimes, which is necessary given the sensitivity of flow type to the raindrop diameter. For the rain droplet number density $N(D_d)$, a recent parameterisation based on Abel and Boutle (2012) (Eq. 3), rather than the Sekhon and Srivastava (1971) model used in the default UKCA-mode BCS scheme, is used in this study (see Section S2 in the Supplement). Using the Abel and Boutle (2012) scheme for the rain droplet number density makes BCS consistent with warm rain assumptions in
the UM. In Eq. 3, $N_0$ and $\lambda$ are the intercept and slope of the rain droplet size distribution, while $R$ is the rain rate in mm hr[-1]. Alternative models for $N(D_d)$ and $U_t(D_d)$ are provided in Sections 2.2-2.3 of Wang *et al.* (2010).

$$N(D_d) = N_0 e^{-kD_d} \qquad \text{(Eq. 3a)}$$
$$N_0 = 4.9 \times 10^7 \, R^{-0.89} \qquad \text{(Eq. 3b)}$$
$$\lambda = 6.236 \times 10^3 \, R^{-0.4} \qquad \text{(Eq. 3c)}$$

The approach of Croft *et al.* (2009, 2010) is adopted to determine number and mass mean scavenging coefficients by integrating $\Lambda(d_p)$ over the aerosol number and mass size distributions (Eqs 4-5).

$$\Lambda_N(\overline{d_p}) = \frac{\int_0^\infty \Lambda(d_p) n(d_p)\, dd_p}{\int_0^\infty n(d_p)\, dd_p} \qquad \text{(Eq. 4)}$$

$$\Lambda_M(\overline{d_p}) = \frac{\int_0^\infty \Lambda(d_p) d_p{}^3 n(d_p)\, dd_p}{\int_0^\infty d_p{}^3 n(d_p)\, dd_p} \qquad \text{(Eq. 5)}$$

In Eqs. 4-5, the size-dependent particle number distribution $n(d_p)$ is modelled assuming a lognormal distribution with the geometric median diameter $(\overline{d_p})$ and the geometric width $(\sigma)$ as parameters. $\Lambda_N(\overline{d_p})$ and $\Lambda_M(\overline{d_p})$ are calculated offline for a
range of $R$, $\overline{d_p}$, and $\sigma$ using Python 3 (Van Rossum and Drake, 2009) scripts. The interpolation points for $R$, $\overline{d_p}$, and $\sigma$ are generated using: $R = 10^{-1 + \frac{1}{7} \times (i-1)}$ mm hr[-1] for $i = 1,..,22$; $d_p = 2 \times 10^{-9 + 0.2 \times (j-1)}$ m for $j = 1,..,22$; and $\sigma = 1 + 0.2k$ for $k = 1,..,5$, and were chosen to balance precision with computational cost.





The resulting $\Lambda_N(\overline{d_p})$ and $\Lambda_M(\overline{d_p})$ arrays are $22 \times 22 \times 5$ in size and are hardcoded into a new Fortran subroutine for the
impaction scavenging of mineral dust by rain in UKCA-mode. The inputs to the subroutine are 3-dimensional fields of the rain
rate, modal geometric median diameters and widths, and modal mass and number concentrations. $\Lambda_N(\overline{d_p})$ and $\Lambda_M(\overline{d_p})$ are
then interpolated for convective and dynamic rain separately wherever the rain rate exceeds zero, using a nearest-neighbour
approach for $\sigma$; log-log (base 10) interpolation for $d_p$; and linear interpolation for $R$. These interpolation methods were
independently selected to reduce the root mean square errors (RMSE) when compared to calculating $\Lambda_N(\overline{d_p})$ and $\Lambda_M(\overline{d_p})$
directly in offline simulations. The interpolated $\Lambda_N(\overline{d_p})$ and $\Lambda_M(\overline{d_p})$ are then used to update the modal number and mass
concentrations using the first order decay equation (Eq. 1), and assuming convective and dynamical grid-cell rain fractions of
0.3 and 1 respectively, in line with other UKCA aerosols. The impaction scavenging of dust by snow is treated using the default
single-moment UKCA scheme (Mann *et al.*, 2010).

The variable of interest in the BCS algorithm (Eqs 1-2) is the collection efficiency $E(d_p, D_d)$ or alternatively the scavenging
coefficient $\Lambda(d_p)$. Various approaches to determine either $E(d_p, D_d)$ or $\Lambda(d_p)$ are outlined below (Sections 2.2-2.6).

### 2.2 Brownian diffusion, interception, and inertial impaction

The classical Slinn (1984) model for the collection efficiency combines what were historically seen as the dominant processes
governing BCS: Brownian diffusion (Eq. 6), interception (Eq. 7), and inertial impaction (Eq. 8). Brownian diffusion efficiently
collects nucleation (diameters of $d_p \leq 0.01$ μm) and Aitken particles that move unpredictably against the air flow around the
raindrop. Inertial impaction collects coarse particles with large mass that are unable to move with the streamlines around the
falling raindrop. Finally, interception occurs when coarse particles are directly within a collection area of the falling raindrop
and is thus independent of the particle's mass or inertia. The Slinn (1984) model has been described in detail by various authors
(e.g., Seinfeld and Pandis, 1998; Berthet *et al.*, 2010; Wang *et al.*, 2010) and is presented in its entirety in Section S3 in the
Supplement. The overall formulae for the individual collection efficiencies are presented in Eqs 6-8 and the reader is referred
to Section S3 and Table S1 in the Supplement for further details of the variables and their dependencies. The dimensionless
parameters in Eqs 6-8 include: $R_{e,r}$ and $R_{e,D}$ are the Reynolds numbers according to raindrop radius and diameter, respectively;
$S_c$ is the Schmidt number; $\phi$ is the ratio of aerosol to raindrop diameter; $S_t$ is the Stokes number; and $S_t^*$ is the critical Stokes
number. $\rho_p$ and $\rho_w$ are respectively the particle density and water density (in kg m$^{-3}$). Salient points of the algorithm include
that an empirical correction factor introduced by Fredericks and Saylor (2016) for the inertial impaction collection efficiency
(Eq. 8) is applied, and all formulae for underlying variables are from Seinfeld and Pandis (1998) except for water viscosity
($\mu_w$, kg m$^{-1}$ s$^{-1}$) which is taken from Dehaoui *et al.* (2015).

$$E_{br}(D_d, d_p) = \frac{4}{R_{e,r}S_c}\left[1 + 0.4R_{e,r}^{\frac{1}{2}}S_c^{\frac{1}{3}} + 0.16R_{e,r}^{\frac{1}{2}}S_c^{\frac{1}{2}}\right] \qquad \text{(Eq. 6)}$$




$$E_{in}(D_d, d_p) = 4\phi \left[ \omega^{-1} + \left(1 + 2R_{e,r}^{\frac{1}{2}}\right)\phi \right] \tag{Eq. 7}$$

$$E_{im}(D_d, d_p) = \begin{cases} \left(\dfrac{S_t - S_t^{*}}{S_t - S_t^{*} + 2/3}\right)^{3/2} \left(\dfrac{\rho_w}{\rho_p}\right)^{1/2} \times & S_t > S_t^{*} \\ 10^{2.905 - 3.07\left(\log_{10}\frac{S_t}{S_t^{*}}\right)^{0.173} - 2.61\times10^{-14}R_{e,D}^{3.9}} & \\ 0 & S_t \leq S_t^{*} \end{cases} \tag{Eq. 8}$$

**2.3 Thermophoresis, diffusiophoresis, and electric charge**

It has long been known that the classical Slinn (1984) model underpredicts the collection efficiency in the accumulation size mode when compared to observations (e.g., Davenport and Peters, 1978). To overcome this deficiency, various other
microphysical processes have been used to explain this disparity including thermophoresis (Eq. 9), diffusiophoresis (Eq. 10), and electric charge effects or 'electrophoresis' (Eq. 11) (Davenport and Peters, 1978; Andronache *et al.*, 2006). Formulae for the individual collision efficiencies are widely published (e.g., Davenport and Peters, 1978; Wang *et al.*, 2010), and the model is described in detail in Sections S4 and S5 in the Supplement, with only formulae for the collection efficiencies presented here (Eqs 9-11). In Eqs 9-11, $\alpha_{th}$, $\beta_{dph}$, and $K$ are empirical scaling factors; $P_r$ is the Prandtl number for air; $T_a$ and $T_s$ are the
temperatures of the air and droplet respectively (in K); $S_{cw}$ is the Schmidt number for water in air; $p_a^o$ and $p_s^o$ are the vapour pressures of water in air at temperatures $T_a$ and $T_s$ respectively (in Pa); $RH$ is the relative humidity (in %); $Q_d$ and $q_p$ are electric charge densities of the droplet and particle respectively (in Coulombs); $C_c(d_p)$ is the Cunningham slip correction factor; and $\mu_a$ is the viscosity of air (in kg m$^{-1}$ s$^{-1}$).

For the purposes of this study, it is assumed that the temperature difference between the air and the raindrop surface ($T_a - T_s$) is 3 K and the electric charge coefficient $\alpha$ used implicitly in Eq. 11 is set to 2, representing standard tropospheric conditions (Wang *et al.*, 2010). Formulae for the water vapour diffusivity in air ($D_{\text{diffwater}}$, m$^2$ s$^{-1}$) and the thermal conductivity of air ($k_a$, J m$^{-1}$ s$^{-1}$ K$^{-1}$) are from Pruppracher and Klett (2010), the thermal conductivity of the particle ($k_p$, J m$^{-1}$ s$^{-1}$ K$^{-1}$) is set to 0.5 following Ladino *et al.* (2011), and an equation for the saturation vapour pressure of water ($p_a^o$ and $p_s^o$) is from Seinfeld and
Pandis (1998).

$$E_{th}(D_d, d_p) = \frac{4\alpha_{th}\left(2 + 0.6\, R_{e,r}^{\frac{1}{2}} P_r^{\frac{1}{3}}\right)(T_a - T_s)}{U_t(D_d)D_d} \tag{Eq. 9}$$

$$E_{df}(D_d) = \frac{4\beta_{dph}\left(2 + 0.6\, R_{e,r}^{\frac{1}{2}} S_{cw}^{\frac{1}{3}}\right)\left(\dfrac{p_s^o}{T_s} - \dfrac{p_a^o RH}{T_a}\right)}{U_t(D_d)D_d} \tag{Eq. 10}$$





$$E_{es}(D_d, d_p) = \frac{16KQ_dq_pC_c(d_p)}{3\pi\mu_aU_t(D_d)D_d{}^2d_p}$$

(Eq. 11)

## 2.4 Rear-capture

Many of the numerical models that were used to develop the semi-empirical relationships between the collection efficiencies and the environmental variables (e.g., Eqs 6-11) made pragmatic assumptions such that the collector raindrop and collected particle were both spherical and that the flow around the raindrop was Stokes or potential flow (e.g., Slinn, 1984). These assumptions are inaccurate for raindrops with diameters $D_d > 280$ µm, wherein the raindrop becomes oblate and is prone to oscillate, and the surrounding flow is viscous and asymmetric (Quérel *et al.*, 2014). Beard and Grover (1974) and Beard (1974) used a complex numerical model with a more accurate representation of the viscous flow around a raindrop than Slinn (1984) to discern the impact of raindrop-induced vortices on the collection efficiency, albeit still assuming both raindrop and particle to be spherical. They found that for intermediate Reynolds numbers ($R_{e,D}$) such that $20 \leq R_{e,D} \leq 400$ (equivalent to $280 \leq D_d \leq 1260$ µm) the rear-capture effect is an important mechanism for aerosol collection. Measurements from Wang and Pruppacher (1977) suggest that for raindrops with $D_d > 1260$ µm the rear capture effect progressively decreases.

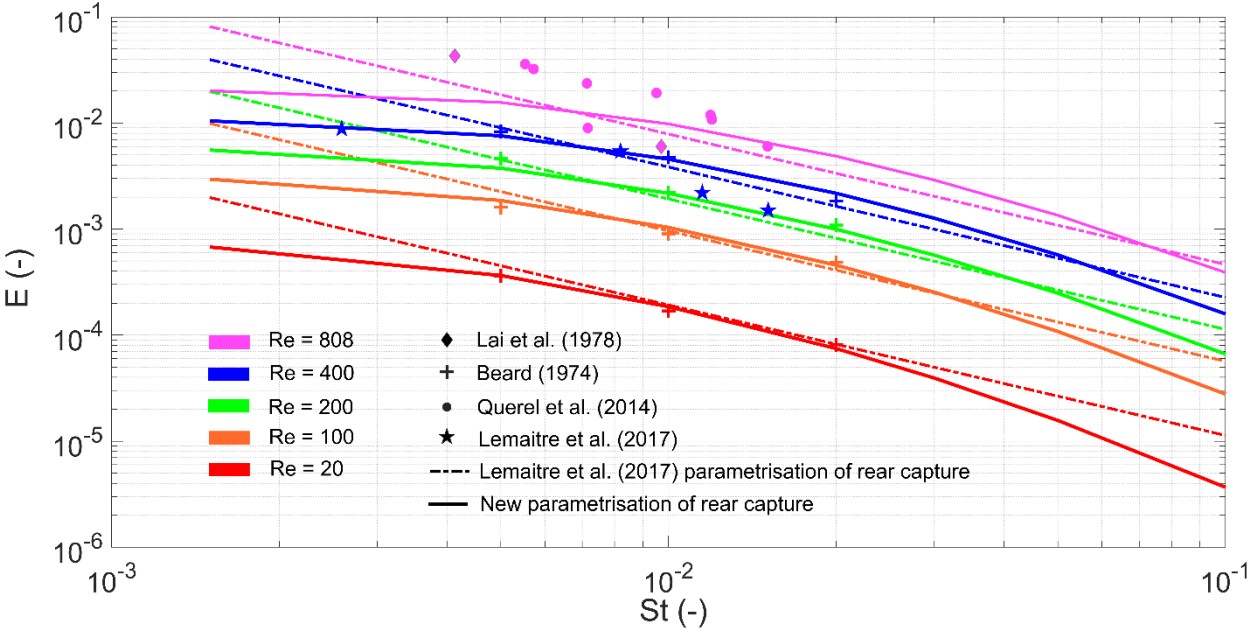

**Figure 1: A new parameterisation of the collision efficiency due to rear-capture in the droplet wake. Also plotted are the data used to fit the parameterisation and the original model of Lemaitre *et al.* (2017)**

Recent laboratory studies by Quérel *et al.* (2014) and Lemaitre *et al.* (2017) have shone light on the importance of the rear-capture effect. By comparing the results of Slinn (1984) and Beard (1974), Lemaitre *et al.* (2017) derived a semi-empirical



formula for the collection efficiency due to rear-capture as a function of Reynolds number, which characterises the flow around the raindrop, and Stokes number ($S_t$), which characterises the particle's inertia and susceptibility to capture. This model was

valid for Reynolds numbers between $20 \leq R_{e,D} \leq 400$ and for Stokes numbers between $5 \times 10^{-3} \leq S_t \leq 5 \times 10^{-2}$ (equivalent to $0.3 \leq d_p \leq 1.1$ µm for $280 \leq D_d \leq 1260$ µm), which is a rather limited subset of the raindrop and particle diameter spectra. Quérel *et al*. (2014) found the rear-capture effect to be important up to a $R_{e,D} \approx 800$ ($D_d \approx 1910$ µm), with the disparity attributed to the assumption of spherical raindrops by Beard (1974). In this paper, a new parameterisation of the collision efficiency via rear-capture is presented – fit to a greater range of observations (Fig. 1) - which is applicable to the

entire aerosol size spectrum (Eq. 12). Crucially, the new collision efficiency asymptotes to zero with decreasing aerosol diameter, following the logic that nanometre-sized aerosols are more likely to be collected by frontal capture via Brownian diffusion. Equation 12 is applicable for $20 \leq R_{e,D} \leq 800$, while for $R_{e,D}$ outside this range it's pragmatically assumed here that $E_{rc}(D_d, d_p) = 0$.

$$E_{rc}(D_d, d_p) = \frac{1}{1.37 \times 10^{10}} S_t^{-3.625} R_{e,D}^{1.444} e^{-0.243 \, (\ln S_t)^2} e^{0.08144 (\ln S_t) \ln R_{e,D}} \qquad (\text{Eq. } 12)$$

### 2.5 Wang *et al*. (2014) model for Λ

Various studies have suggested that the disparity between observed and modelled BCS rates is mostly attributable to confounding atmospheric processes such as nucleation scavenging, turbulent diffusion, and precipitation-induced downdrafts (e.g., Wang *et al*., 2010, 2011; Andronache *et al*., 2006). Wang *et al*. (2010) in particular recommend that the theoretical BCS

models with the greatest values of Λ should be used in GCMs. Given the complexity of such schemes (e.g., Eqs 6-12 and Sections S1-S5 in the Supplement), it is useful to derive simplified formulae that can reduce the computational cost of explicitly calculating Λ online in a GCM. In answer to this, Wang *et al*. (2014) fit a simple polynomial formula to the upper 90th percentile of Λ from various theoretical models in the literature as a function of aerosol diameter and rain rate (Eq. 13), that can be used instead of explicitly evaluating Λ using Eq. 2. The coefficients in Eq. 13 are provided in Table 8 in Wang *et al*.

(2014) and Table S2 in the Supplement. In Eq. 13, $d_p$ is in units of µm rather than units of metres used elsewhere in this study.

$$\Lambda(d_p) = A(d_p) R^{B(d_p)} \qquad (\text{Eq. } 13a)$$

$$\log_{10} A(d_p) = \begin{cases} \sum_{i=0}^{3} a_i (\log_{10} d_p)^i & d_p \leq 2 \text{ µm} \\ \sum_{i=0}^{6} b_i (\log_{10} d_p)^i & d_p > 2 \text{ µm} \end{cases} \qquad (\text{Eq. } 13b)$$





$$B(d_p) = \begin{cases} \sum\limits_{i=0}^{1} c_i \left(\log_{10} d_p\right)^i & d_p \leq 2 \text{ μm} \\ \sum\limits_{i=0}^{6} e_i \left(\log_{10} d_p\right)^i & d_p > 2 \text{ μm} \end{cases} \qquad (\text{Eq. } 13c)$$

### 2.6 Laakso *et al.* (2003) model for $\Lambda$

Laakso *et al.* (2003) derived a formula for $\Lambda$ as a function of aerosol diameter and rainfall intensity, using 6 years of measurements from a boreal forest site in Southern Finland (Eq. 14). Their model is widely used in GCMs but was only fit to a limited range of $R$ and $d_p$: $R \leq 20$ mm hr$^{-1}$ and $0.01 < d_p < 0.5$ μm. However, Fig. 7 in Laakso *et al.* (2003) shows that the model does an excellent job at capturing observed $\Lambda$ from Volken and Schumann (1993) for $0.5 < d_p < 10$ μm, albeit for a single value of $R$. Given the strong gradient in $\Lambda$ with $d_p$ at $d_p = 0.5$ μm, it seems appropriate to extend this model up to 10 μm with the necessary caveats attached. Outside these range of values (i.e., for $R > 20$ mm hr$^{-1}$, $d_p < 0.01$ μm, and $d_p > 10$ μm) the values at the extrema are used. As with Wang *et al.*'s (2014) model for $\Lambda$, Eq. 14 can be used instead of explicitly evaluating Eq. 2 in the algorithm described in Section 2.1. In Eq. 14, $d_p$ is in units of metres, and the coefficients $a_i$ are $a_0 = 274.35758$, $a_1 = 332839.59273$, $a_2 = 226656.57259$, $a_3 = 58005.91340$, $a_4 = 6588.38582$, and $a_5 = 0.244984$.

$$\Lambda(d_p) = 10^{A(d_p)} \qquad (\text{Eq. } 14a)$$

$$A(d_p) = \sum_{i=0}^{4} a_i \left(\log_{10} d_p\right)^{-i} + a_5 R^{0.5} \qquad (\text{Eq. } 14b)$$

### 3 Box-model simulation design

The BCS algorithm as described in Section 2, with $\Lambda_N\left(\overline{d_p}\right)$ and $\Lambda_M\left(\overline{d_p}\right)$ tabulated assuming various collision efficiencies and BCS rates (Sections 2.2-2.6), was first tested in offline box model simulations before being implemented in the UM. The box model simulations use a simple forward Euler time stepping scheme, with 1 minute time increments and 180 timesteps (or 3 hours total duration). Three different rain rates were tested corresponding to drizzle ($R = 0.5$ mm hr$^{-1}$), moderate rain ($R = 2.5$ mm hr$^{-1}$), and heavy rain ($R = 10$ mm hr$^{-1}$). Two initial condition (IC) size distributions are tested: an accumulation mode with ICs of $\overline{d_p} = 0.4$ μm and $\sigma = 1.59$, and a coarse mode with ICs of $\overline{d_p} = 2$ μm and $\sigma = 2$. The initial IC distributions are arbitrary and intended to represent standard dust conditions in the accumulation and coarse regimes. The results of the box-model simulations and direct comparisons between the scavenging coefficients and collision efficiencies are provided in Sections 5.1-5.2.


The GLOMAP-mode aerosol model was originally developed as a bin scheme (GLOMAP-bin, Spracklen *et al.*, 2005), with
20 logarithmically spaced size bins spanning 2 nm to 22 μm. In order to test the impact of BCS on the modal width ($\sigma$), which
relates to KQ4, the offline box model is further run with the GLOMAP-bin size bins, extended upwards by 4 bins to 150 μm.
Specifically, the lognormal cumulative distribution function is used to obtain the initial mass and number concentrations in
each bin for an initial lognormal distribution. The box model is then integrated over each bin individually, using the geometric
mean of the bin thresholds as a representative diameter and the SLINN+PH+RC BCS coefficients to determine the scavenging
rate per bin. Finally, lognormal distributions are fit to the bins at T+1H (1 hour elapsed) and T+3H (3 hours elapsed) by
generating random variables (RV) from the histograms in Python 3 (Van Rossum and Drake, 2009) and fitting a lognormal
distribution to the RVs using Maximum Likelihood Estimation. A comparison of BCS applied to a bin aerosol model and to a
modal aerosol model is provided in Section 5.2.

## 4 The Met Office Unified Model configuration and simulation design

### 4.1 UM configuration (UM-GA8.0)

In order to compare the various BCS schemes outlined in Section 2, GCM simulations were performed using the Met Office
UM in an atmosphere-only mode with the latest science configurations Global Atmosphere vn8 (GA8.0) and Global Land
vn9.0 (GL9.0). A technical overview of GA8.0/GL9.0 has not yet been published, but in effect GA8.0/GL9.0 consolidates the
changes introduced at GA7.1 (Walters *et al.*, 2019) including the introduction of a cloud droplet spectral dispersion
parameterisation based on Liu *et al.* (2008), near-surface drag improvements (Williams *et al.*, 2020), and multiplicative scaling
of DMS emissions (Bodas-Salcedo *et al.*, 2019). Although the UM can be run at various resolutions, the resolution used here
is the climate configuration N96L85, i.e., 1.875º longitude by 1.25º latitude with 85 vertical levels up to a model lid at 80 km,
with 50 levels below 18 km altitude, and a model timestep of 20 mins (Walters *et al*, 2019). The model is technically named
after its science configuration (UM-GA8.0) which we adopt in this study.


UM-GA8.0 includes the coupled UKCA-mode aerosol and chemistry scheme which holistically simulates atmospheric
composition in the Earth System, with chemistry called once per model hour at N96L85 and emissions updated every timestep
(Archibald *et al.*, 2020). UM-GA8.0 uses a simplified UKCA chemistry configuration, with important oxidants ($O_3$, OH, $NO_3$,
$HO_2$) prescribed as monthly mean climatologies (Walters *et al.*, 2019; Mulcahy *et al.*, 2020). UKCA-mode includes a
prognostic double-moment aerosol scheme that carries aerosol mass and number concentrations in a predetermined number of
log normal modes spanning nucleation to coarse sizes (Mann *et al.*, 2010; Mulcahy *et al.*, 2020). In its default configuration,
UKCA-mode comprises 4 soluble modes (nucleation, Aitken, accumulation, and coarse), as well an insoluble Aitken mode,
with 4 aerosol constituents represented: sulphate ($SO_4$), Sea-Salt (SS), Black Carbon (BC), and Organic Carbon (OC).
Although UM-GA8.0 incorporates the CLASSIC mineral dust scheme by default (Woodward, 2001), we elect to use the inbuilt
UKCA-mode dust scheme in this investigation, which comprises externally mixed dust in 2 insoluble modes (Section 4.2).



The direct aerosol radiative effect is treated with UKCA-Radaer, which uses look-up tables of Mie extinction parameters based on size and a volume-mixed refractive index based on speciated ambient aerosol concentrations (Bellouin *et al.*, 2013). Aerosol water content and hygroscopic growth of the soluble modes is simulated prognostically using the Zdanovskii-Stokes-Robinson

(ZSR) method.

## 4.2 UKCA-mode dust and dust emissions scheme

The UKCA-mode dust scheme is mostly unchanged from Mann *et al*. (2010). Mineral dust is represented by accumulation and coarse insoluble modes with fixed geometric widths of 1.59 and 2 respectively. Functionality exists in UKCA-mode to permit dust ageing into the equivalent soluble modes, from acting as condensation nuclei for soluble vapours or by coagulation with

soluble aerosols, but at the present time these processes are not included in our simulations and dust remains insoluble throughout its atmospheric lifetime. Owing to the assumption of insolubility, dust is not permitted to act as liquid cloud condensation nuclei (CCN) and thus be removed from the atmosphere by nucleation scavenging in these simulations. This is a simplification, as insoluble aerosol can act as CCN according to Köhler theory, albeit at higher relative humidities than for soluble aerosol (Seinfeld and Pandis, 1998).


Dust emissions are determined each timestep using a method based on the widely used scheme of Marticorena and Bergametti (1995).  Horizontal flux is calculated in nine bins with boundaries at 0.0632, 0.2, 0.632, 2.0, 6.32., 20.0, 63.2, 200.0, 632.0 and 2000.0 μm diameter.  Total vertical flux in six bins up to 63.2 μm is derived from total horizontal flux and follows the size distribution of the horizontal flux in bins 1 to 6. The dry threshold friction velocity for each bin is taken from Bagnold (1941),

while the effect of soil moisture on emissions is treated according to Fécan *et al.* (1999). Further detail on the dust emissions scheme is provided in Woodward *et al.* (2022). Mapping the binned emissions to the UKCA-mode dust scheme requires a degree of pragmatism and trial-and-error. In previous testbed simulations, an optimal mapping emerged wherein Bin 2 + ½ Bin 3 was emitted to the accumulation mode while ½ Bin 3 + Bin 4 + Bin 5 were emitted to the coarse mode (Jones *et al*., 2021). This mapping is subject to change given ongoing improvements to the dust scheme. Note that both Bin 1 (0.0632 <

$d_p$ < 0.2 μm) and Bin 6 (20 < $d_p$ < 63.2 μm) emissions, which are included in CLASSIC, are missing from UKCA-mode dust, which constitutes a large fraction of the total particle number (Bin 1) and mass (Bin 6) emitted. In future, a third insoluble mode representing giant dust particles (e.g., Ryder *et al*., 2019) may be added to UKCA-mode to increase the degrees of freedom to 5 in line with CLASSIC and further resolve the span of the emitted dust size distribution, but that is outside the scope of this work.


The density of mineral dust is assumed to be invariant at 2650 kg m$^{-3}$ (Mahowald *et al*., 2014), with refractive indices from Balkanski *et al*. (2007). Dry deposition and sedimentation in UKCA-mode follow the double-moment resistance type framework outlined by (Mann *et al*., 2010) with sub-timesteps of 30 mins and 15 mins for the accumulation and coarse





insoluble modes respectively. Downward mode merging (i.e., KQ4, see the Introduction) follows the approach outlined in

Mann *et al.* (2010) and is initiated when the coarse mode median diameter falls below the critical threshold of $d_p = 1\,\mu m$, whereupon mass and number are transferred from the coarse insoluble mode to the accumulation insoluble mode. The maximum fraction of the initial number and mass concentration permitted to be transferred per time step is 50 % and 99 % respectively, following UKCA-mode's existing mode-merging protocol. Note that only a subset of simulations include mode-merging (see Table 1).

**4.3 UM-GA8.0 simulation design**

UM-GA8.0 simulations are performed using standard Atmospheric Model Intercomparison Project (AMIP) protocol. UM-GA8.0 uses CMIP6-defined historical greenhouse gas and aerosol emissions and concentrations fields as detailed by Sellar *et al.* (2020). Sea-surface temperature and sea-ice fields are fixed timeseries from the NOAA high-resolution blended analysis of daily SST and ICE (OISSTV2) reanalysis product (Reynolds *et al.*, 2007) and are updated daily. The simulations are run for

20 model years (1989-2008), with atmospheric mineral dust concentrations initialised to zero. Given the spin-up time necessary for atmospheric dust concentrations to reach equilibrium, only the last 15 model years are used for the analysis.

| Simulation name | Dust and BCS scheme description | KQ(s) |
|---|---|---|
| SLINN | UKCA 2-mode dust scheme with Slinn (1984) collection efficiencies | 2 |
| SLINN+PH | Same as SLINN but with phoresis and charge collection efficiencies added | 2 |
| SLINN+PH+RC | Same as SLINN+PH but with rear-capture collection efficiency added | 1, 2, 3, 4 |
| WANG | UKCA 2-mode dust scheme with BCS following Wang *et al.* (2014) | 1 |
| LAAKSO | UKCA 2-mode dust scheme with BCS following Laakso *et al.* (2003) | 1, 4 |
| SLINN+PH+RC(1M) | Same as SLINN+PH+RC but a single moment scheme with the modal median diameters used to interpolate the scavenging coefficient and no consideration of mode widths | 3 |
| SLINN+PH+RC(DM) | Same as SLINN+PH+RC but with downward mode merging applied to the coarse insoluble mode | 4 |
| LAAKSO(DM) | Same as LAAKSO but with downward mode merging applied to the coarse insoluble mode | 4 |

**Table 1. Description of the UM-GA8.0 simulations performed, and the key questions (KQs) addressed by each simulation**

Table 1 describes the simulations performed for this study, including which Key Questions or KQs (see Introduction) are pertinent to each simulation. Note that the same nomenclature is adopted for the offline BCS model and box model as for the





name of the corresponding UM-GA8.0 simulations except with lowercase and italics. For example, *Slinn* refers to the BCS
scheme proposed by Slinn (1984) (Section 2.2) while SLINN refers to the UM-GA8.0 simulation which employs the *Slinn*
BCS model. In addition to testing the various double-moment BCS approaches (Section 2.2-2.6), we additionally test the
assumption of a single-moment BCS scheme using the *Slinn+ph+rc* model for $\Lambda$ in simulation SLINN+PH+RC(1M), and the
impact of including downward mode merging in theoretical and empirical BCS models in SLINN+PH+RC(DM) and
LAAKSO(DM) respectively. Note that *Slinn+ph+rc* is the default model used as the basis for answering KQ3 and KQ4, as
well as representing theoretical schemes in answering KQ1. The reason *Slinn+ph+rc* was chosen rather than *Slinn*, *Slinn+ph*,
*Wang*, or *Laakso* is that it fulfils Wang *et al.* (2010)'s recommendation that the best BCS model to use is the theoretical scheme
with the highest BCS rates (see Section 5.1). A working hypothesis is then that *Slinn+ph+rc* most accurately reflects the real-
life BCS process of the models tested.

**4.4 Validatory observations**

A range of observations are employed to test the fidelity of the individual BCS schemes. For seasonal dust optical depth (DOD)
at 440 nm, observations are provided by the Aerosol Robotic Network (AERONET, Holben *et al.*, 1998) at 8 'dusty' locations
from those selected by Bellouin *et al.* (2005). Also, we use regional 550 nm DODs as synthesised from models and observations
by Kok *et al.* (2021), based on Ridley *et al.* (2016) observations for the Northern Hemisphere and Adebiyi *et al.* (2020) for the
Southern Hemisphere. The criteria imposed for selecting 'dusty' AERONET stations is at least 4 years of continuous monthly
data with at least 10 daily means per month, and an aerosol Angstrom exponent (870-440 nm) below 0.5 for at least 10 months
of the year. For near-surface dust concentrations, we employ seasonal-mean observations from the historical University of
Miami Oceanic Aerosols Network (U-MIAMI) (Prospero and Nees, 1986) which is often used to validate dust models (e.g.,
Peng *et al.*, 2012; Checa-Garcia *et al.*, 2021). A global network of dust deposition fluxes is provided by Huneeus *et al.* (2011).
The AERONET DODs and U-MIAMI observations are presented in Tables S3 and S4 in the Supplement respectively, whilst
the Kok *et al.* (2021) DODs and Huneeus *et al.* (2011) deposition rates are available from their respective papers.

For the dust particle size distributions (PSD), which are used to evaluate the impact of representing downward mode merging
(Section 5.6), observations are compiled from a transatlantic transect of 3 independent aircraft campaigns: Fennec 2011
representing dust near the source regions of Mali and Mauritania (Ryder *et al.*, 2013), AER-D representing dust in the Saharan
Air Layer (SAL) over the east equatorial Atlantic (Ryder *et al.*, 2018), and the Saharan Aerosol Long-Range Transport and
Aerosol–Cloud-Interaction Experiment (SALTRACE) campaign representing dust over the west equatorial Atlantic (Weinzierl
*et al.*, 2017) , with additional processing as described in Ryder *et al.* (2019).. We use the campaign mean fitted PSDs presented
in Fig. 9 of Ryder *et al.* (2019) for Fennec 2011 and AER-D, which each comprise a quadrimodal lognormal size distribution
with 10[th] and 90[th] percentiles. For SALTRACE, we use number size distributions (NSDs) and volume size distributions (VSDs)
collected from a single straight and level run during SALTRACE flight 130622a (22[nd] June 2013), alongside 16% and 84%
percentiles. The PSDs were inferred using the Bayesian inversion algorithm of Walser *et al.* (2017). The following averaging





regions are used to approximately collocate simulated dust concentrations with the observations: (4-8 °W, 21-26 °N) and 0.1-1.2 km altitude for Fennec 2011 to coincide with the fresh dust observations, (18-24 °W, 14-24 °N) and 2-3 km altitude for AER-D, and (58-61 °W, 11-14 °N) and 2-2.4 km altitude for SALTRACE. Temporally, Fennec 2011 and SALTRACE are

taken to represent conditions in June and AER-D in August, i.e., the month of operation for each campaign.

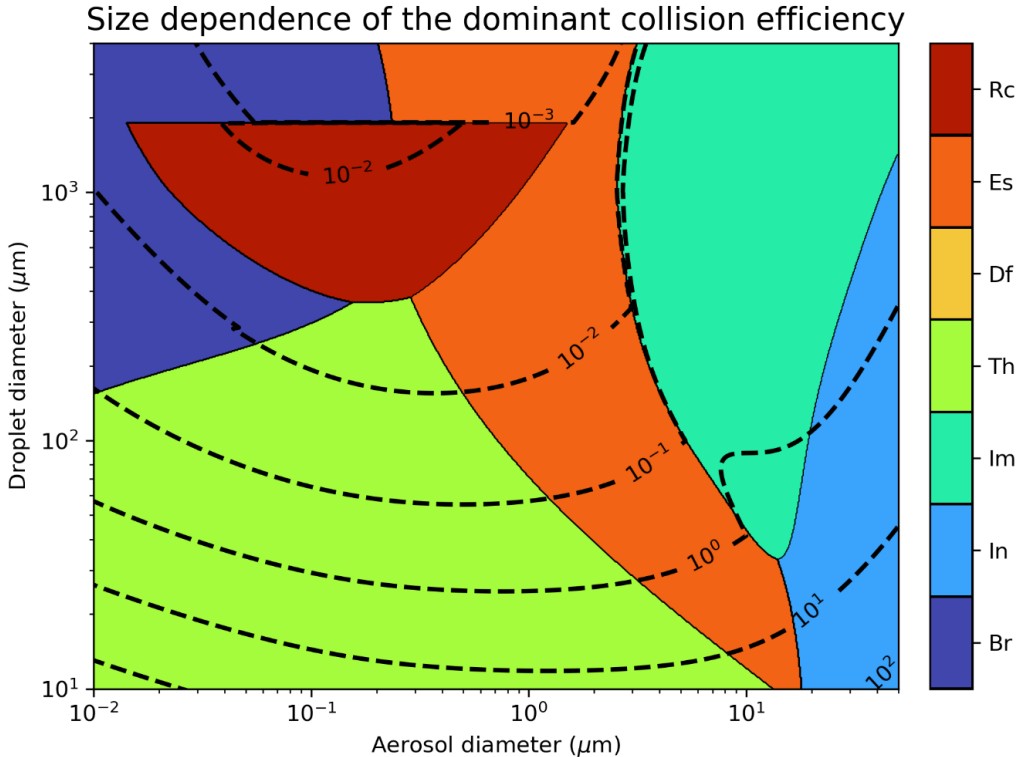

**Figure 2. The dominant contributor to the total collision efficiency as a function of aerosol diameter and rain droplet diameter, where Rc = rear-capture, Es = electric charge, Df = diffusiophoresis, Th = thermophoresis, Im = inertial impaction, In =**
**interception, and Br = Brownian diffusion. Dashed lines show logarithmically spaced contours of total collision efficiency**

## 5 Results

### 5.1 Collision efficiencies and scavenging coefficients

Before comparing the BCS schemes in the UM-GA8.0 simulations, it is useful to directly compare collision efficiencies and scavenging rates between the models. Given that a new formulation for the 'rear-capture' collision efficiency is provided in

this paper (Eq. 12), it is also useful to assess if and when rear-capture makes an important contribution to the overall collision efficiency. Figure 2 shows the dominant collision efficiency as a function of aerosol diameter and rain droplet diameter, for the processes outlined in Sections 2.2-2.4. It is clear that rear-capture (Rc) makes a substantial contribution to the overall collision efficiency for a large portion of the aerosol and raindrop size spectrum, in particular, in the Greenfield gap for



accumulation sized aerosols and moderate to large rain droplet diameters (400 μm – 2 mm). Figure 2 also highlights that the
Slinn (1984) processes of Brownian diffusion (Br), interception (In), and impaction (Im) only dominate the collision efficiency
for a limited size subspace. For aerosol diameters between 0.2 μm – 3 μm, rear-capture, thermophoresis (Th), and electric
charge (Es) are consistently the dominant BCS processes.

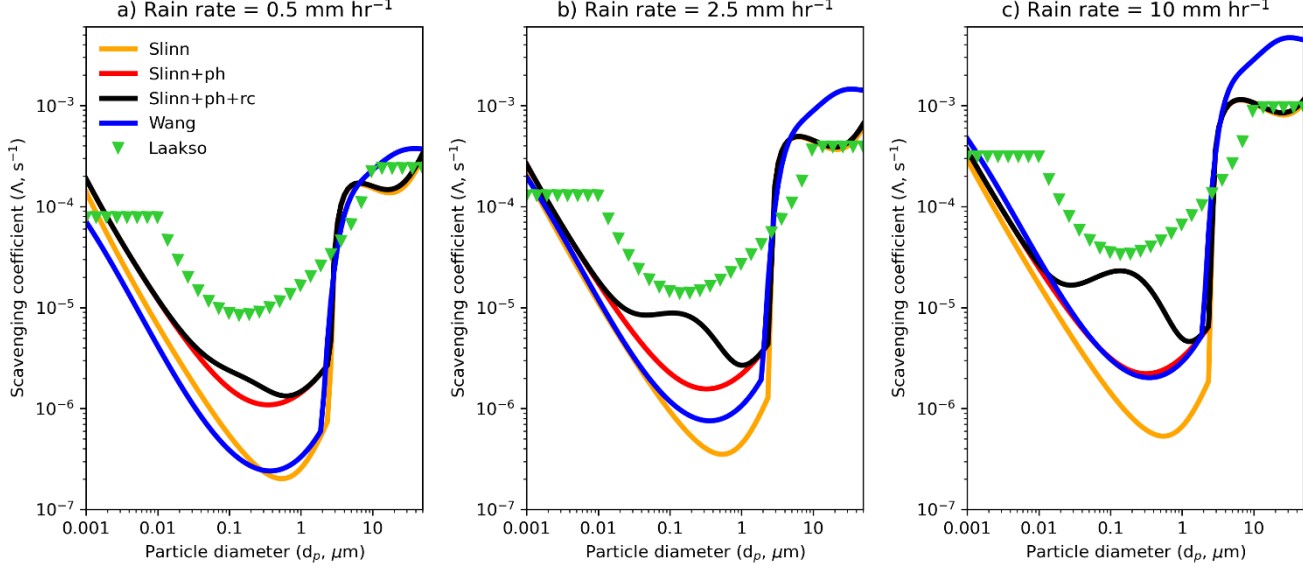

**Figure 3. BCS scavenging coefficient (Λ, Eq. 2) as a function of aerosol diameter for 5 BCS models (Section 2), and for rain rates
representing (a) drizzle, (b) moderate rain, and (c) heavy rain**

Figure 3 shows the BCS rate as a function of aerosol diameter for the BCS models outlined in Section 2, and for 3 rain rates
corresponding to (a) drizzle, (b) moderate rain, and (c) heavy rain. It is clear that in the Greenfield gap the empirically derived
Λ (i.e., *Laakso*) is markedly greater than the theoretical Λ, for example, being an order of magnitude greater than *Slinn+ph+rc*
at $d_p$ = 1 μm for all 3 rain scenarios. It is also clear from comparing *Slinn* with *Slinn+ph* and *Slinn+ph+rc* that phoresis
significantly enhances Λ for aerosol with diameters less than ~2 μm, while rear-capture has a significant impact in the
Greenfield gap for moderate and heavy rain scenarios. For super coarse aerosol with $d_p$ > 10 μm all BCS schemes exhibit Λ
of the order $1\times10^{-4}$ s$^{-1}$ for drizzle, while the semi-theoretical *Wang* scheme exhibits greater Λ for heavy rain ($4\times10^{-3}$ s$^{-1}$) than
the other models. In general, the *Wang* BCS rates are similar to *Slinn* for drizzle, and between the *Slinn* and *Slinn+ph* rates for
moderate rain, which is surprising given that the *Wang* model was fit to the upper 90th percentile of the existing theoretical
BCS rates and thus should be closer to *Slinn+ph* over the entire rain rate spectrum (Wang *et al*., 2014). Although Fig. 3 shows
Λ computed using atmospheric properties representative of surface conditions ($P$ = 101,325 Pa, $T$ = 20 °C, $RH$ = 80 %),


we find that using standard atmospheric conditions at 5 km altitude only changes Λ by a factor of 2 at most for *Slinn+ph+rc*

and is thus a second-order impact compared to the deviation in Λ with particle diameter (Fig. S4 in the Supplement).

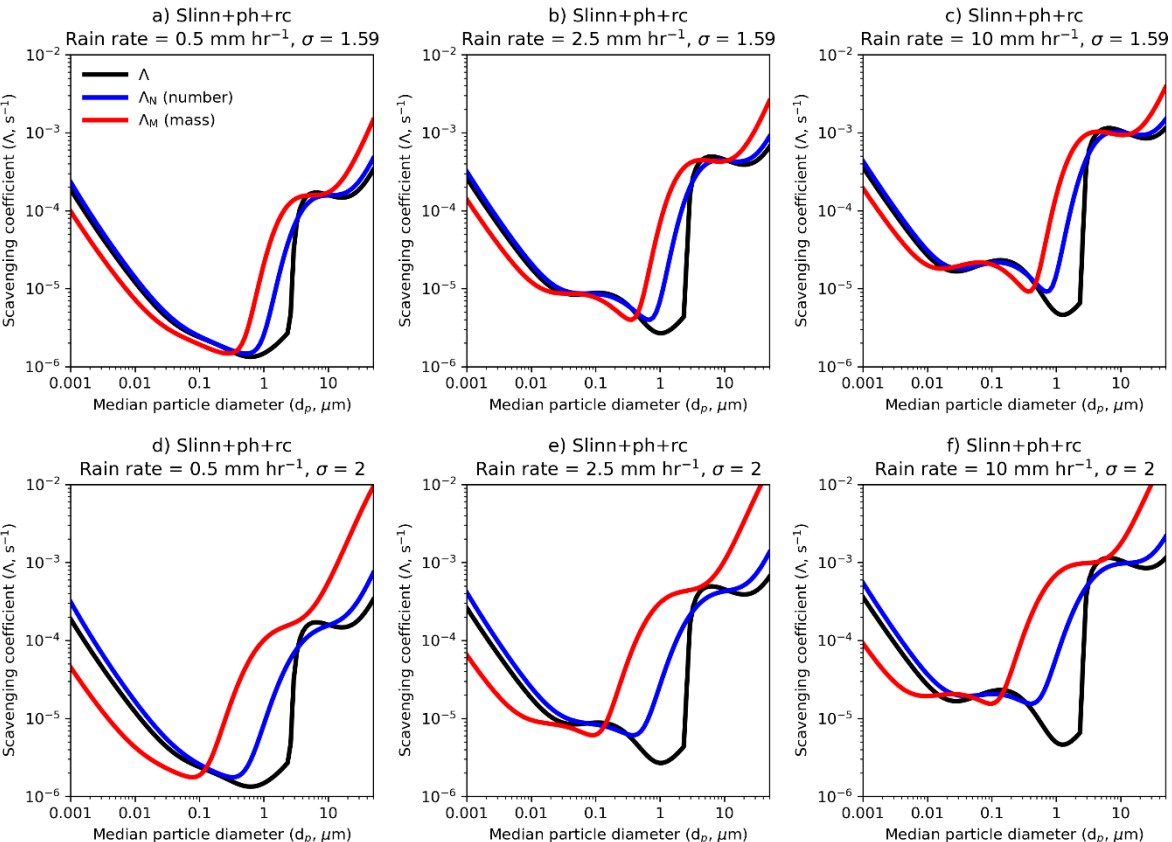

**Figure 4. BCS scavenging coefficient integrated over aerosol mass and number (Eqs 4-5) for geometric widths (a-c) σ = 1.59 and (d-f) σ = 2, as a function of aerosol diameter for rain rates representing (a,d) drizzle, (b,e) moderate rain, and (c,f) heavy rain**


BCS is highly sensitive to aerosol particle size, as shown in Figure 3. This means that a single-moment BCS scheme which applies the same scavenging coefficient to aerosol number and mass concentrations in a mode, or that does not account for the modal width, may be erroneously simplistic. A single-moment BCS scheme is utilised by UKCA-mode in the UM (Mann *et al.*, 2010) and such is the motivation for KQ3. Figure 4 shows the *Slinn+ph+rc* BCS rates for a uniform size distribution (i.e.,

with no integration over the aerosol size distribution, Eq. 2) and for equivalent number and mass distributions (Eqs 3 and 4 respectively) assuming geometric widths of $\sigma = 1.59$ and $\sigma = 2$, representing the accumulation and coarse insoluble modes in UKCA-mode, respectively. It is clear that the number ($\Lambda_N$) and mass ($\Lambda_M$) scavenging coefficients are significantly greater than the uniform scavenging coefficient ($\Lambda$), particularly for $\overline{d_p} > 0.15$ μm and $\sigma = 2$. For example, at $\overline{d_p} \approx 1$ μm, $\Lambda_M$ is a



factor of 150 greater than $\Lambda$ for all 3 rain scenarios for $\sigma = 2$. Additionally, $\Lambda_N$ and $\Lambda_M$ are significantly divergent such that

mass is removed faster than number for $\overline{d_p} > 0.15$ μm but slower than number for $\overline{d_p} < 0.15$ μm suggesting that, if unaffected

by other processes, the aerosol median diameter would converge upon $\overline{d_p} \approx 0.15$ μm for $\sigma = 2$ (Figs 4d-f). For $\sigma = 1.59$, $\Lambda_N$

and $\Lambda_M$ are closer to $\Lambda$ and the aerosol median diameter would converge to $\overline{d_p} \approx 0.4$ μm over time, without accounting for

other sink and source processes (Figs 4a-c).

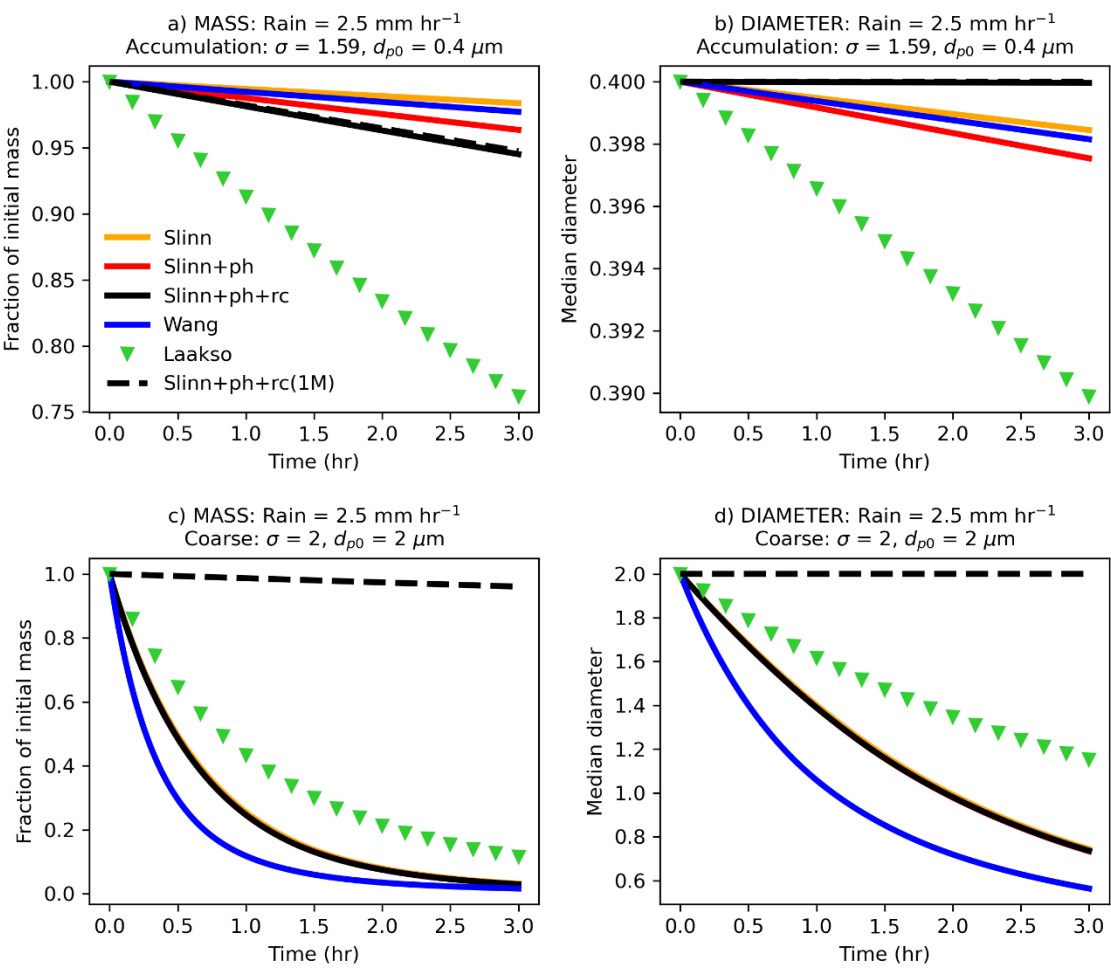


**Figure 5. Time evolution of the (a,c) mass concentration and (b,d) median diameter of (a-b) an accumulation-like mode and (c-d) a coarse-like mode with a constant rain rate of 2.5 mm hr⁻¹ for 6 BCS schemes (see Table 1 for definitions). Results from offline box model simulations**



## 5.2 Box-model results

Before comparing the BCS schemes in a GCM environment, it is useful to compare them in a simple offline box model. Figure 5 shows the time evolution of mass and diameter from box model simulations with each of the BCS schemes, assuming a constant rain rate of 2.5 mm hr$^{-1}$ (results for rain rates of 0.5- and 10-mm hr$^{-1}$ are shown in Figure S5 in the Supplement), and for accumulation and coarse aerosol size modes. Note that all results presented in the section are sensitive to the initial conditions for the 2 modes, and different initial conditions may produce markedly different results given the wide range of $\Lambda$,

$\Lambda_N$, and $\Lambda_M$ (Figs 3-4). It is clear that for these initial conditions there is little deviation in median diameter for the accumulation mode (Fig. 5b) over the 3 model hours for any BCS scheme. However, 2 % of accumulation mode mass is removed by the end of the simulation in *Slinn*, compared to 4 % in *Slinn+ph* and 6 % in *Slinn+ph+rc* (Fig. 5a), which shows that there is some sensitivity to the additional processes missing in *Slinn* (KQ2). These differences are small compared to the *Laakso* scheme which exhibits a 24 % decrease in accumulation mode mass over the 3-hour duration (KQ1).


For the coarse mode, 97 % of mass is removed over the course of 3 hours in the 2-moment *Slinn* and *Wang* models, and 88 % of mass is removed in *Laakso* (Fig. 5c). Additionally, the median diameter evolves from $\overline{d_p} = 2$ μm at the start of the simulation to approximately $\overline{d_p} = 0.75$ μm in the 2-moment *Slinn* models, $\overline{d_p} = 0.56$ μm in *Wang*, and $\overline{d_p} = 1.15$ μm in *Laakso* (Fig. 5d). Figures 5c and 5d also show the significant impact of using a single moment impaction scavenging scheme,

notably that without consideration for the mode width or for the time evolution of $\overline{d_p}$, only 4 % of coarse mode mass is removed in the single moment *Slinn+ph+rc(1M)* model compared to 97 % in *Slinn+ph+rc* (Fig. 5c) (KQ3). The difference in mass evolution between *Slinn+ph+rc* and *Slinn+ph+rc(1M)* can be attributed to the large difference in mass and uniform scavenging coefficients for $\overline{d_p} \approx 2$ μm (Fig. 4e). For the accumulation mode, the mass and uniform scavenging coefficients are similar for $\overline{d_p} = 0.4$ μm in Fig. 4b, hence explaining why there is little difference in accumulation mode mass evolution

between *Slinn+ph+rc* and *Slinn+ph+rc(1M)* in the box model simulations (Fig 5a). Differences between the single-moment and double-moment approaches are explored in Section 5.5 using UM-GA8.0.

The BCS results shown so far have assumed a fixed width for the aerosol size distribution, in line with double-moment modal models that are widely employed in GCMs (Gliß *et al.*, 2021). A more advanced but computationally expensive approach is

to apportion aerosol mass or number into several fixed size bins, which increases the degrees of freedom and allows the width of the aerosol mode to evolve. The BCS box model has also been applied to a bin aerosol scheme (see Section 3 for Methods), assuming the same initial conditions as for the modal aerosol scheme, with the time evolution of the aerosol number density as a function of aerosol diameter shown in Fig. 6. It is clear from Fig. 6c that aerosol number is more efficiently removed from the coarse bins ($d_p > 2$ μm) than for the smaller bins ($d_p < 2$ μm) when the initial conditions are $\overline{d_p} = 2$ μm and $\sigma = 2$, and

thus that the effective width of the binned model decreases to $\sigma = 1.69$ over the course of the 3-hour simulation. Conversely, the width is not permitted to shrink in the modal model, and thus the particle number density ($dN/d\log(d_p)$) for $d_p < 2$ μm



is artificially enhanced by the end of the simulation (Fig. 6d). One potential way to compensate for this effect is to introduce downward mode merging (KQ4), whereupon dust mass and number are moved from the broad coarse mode to the narrow accumulation mode following BCS if the new coarse mode median diameter descends below a threshold value. Downward

merging is explored in Section 4.6 using UM-GA8.0 with the critical threshold value chosen to be 1 μm.

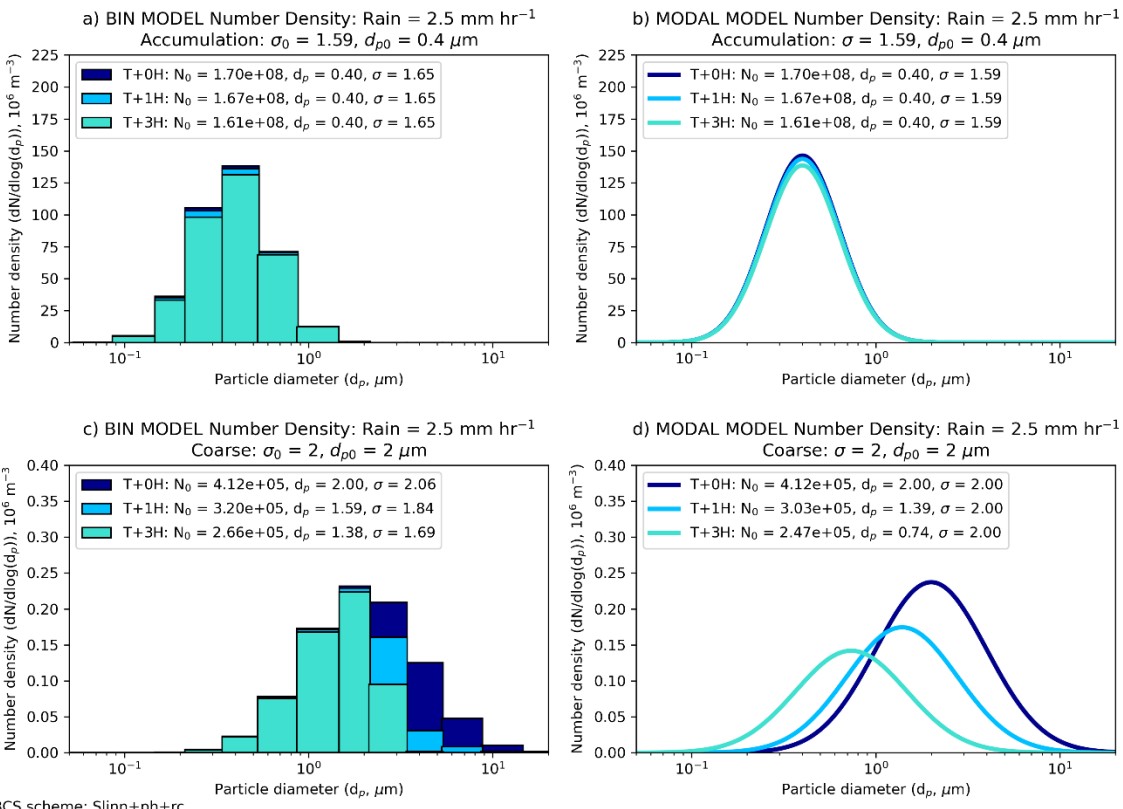

**Figure 6. Number density as a function of particle diameter in (a,c) 'bin' and (b,d) 'modal' simulations with the offline box model. The BCS scheme is *Slinn+ph+rc*, the rain rate is 2.5 mm hr⁻¹ and results are shown for (a-b) an accumulation size distribution and (c-d) a coarse size distribution. The key shows approximate lognormal distributions at T+0H, T+1H and T+3H time intervals**

### 5.3 KQ1: Empirical vs theoretical BCS schemes

We now move to comparing the BCS schemes in the UM-GA8.0 simulations and thus answering the KQs posed in the Introduction. In order to answer KQ1, the SLINN+PH+RC, WANG and LAAKSO simulations are compared in terms of global

dust metrics. SLINN+PH+RC is chosen over SLINN and SLINN+PH to represent a simulation with a theoretical BCS scheme as it resolves more BCS processes. WANG can be thought of as representing a semi-empirical BCS scheme and has the advantage that it much simpler to compute BCS rates using the *Wang* model than the *Slinn+ph+rc* model. LAAKSO represents





an entirely empirical BCS scheme. Table 2 shows global dust metrics from all of the UM-GA8.0 simulations performed in this study. From Table 2, it is clear that the order of magnitude difference between the empirical (LAAKSO) and theoretical

(SLINN+PH+RC) BCS rates for accumulation sized aerosol has significant impact on the global dust mass burden. For example, the global accumulation dust burden in LAAKSO is 1.11 Tg, while in SLINN+PH+RC it is 4.61 Tg, and in WANG it is 8.24 Tg. It is clear that BCS is significantly greater in LAAKSO, with 89 % of accumulation dust removed by wet deposition compared to only 72 % in SLINN+PH+RC and 52 % in WANG. Interestingly, accumulation dust emissions are also enhanced by 5 % in LAAKSO compared to the other models, which can only emanate from a change to meteorology,

either in terms of soil moisture, surface roughness, or near-surface wind speeds in the dust source regions.

|  |  | Dust surface emissions | Global dust burden | Dust surface concentration | Wet deposition fraction | 550nm dust optical depth (DOD) | Dust lifetime |
|  |  | Tg yr$^{-1}$ | Tg | µg m$^{-3}$ | % |  | days |
| --- | --- | --- | --- | --- | --- | --- | --- |
| Accumulation mode | SLINN | 69.2 | 8.42 | 2.11 | 52 | 0.043 | 43.8 |
|  | SLINN+PH | 68.3 | 4.71 | 1.29 | 71 | 0.024 | 24.8 |
|  | SLINN+PH+RC | 69.8 | 4.61 | 1.28 | 72 | 0.024 | 23.8 |
|  | WANG | 68.7 | 8.24 | 2.1 | 52 | 0.042 | 43.1 |
|  | LAAKSO | 73.2 | 1.11 | 0.51 | 89 | 0.006 | 5.4 |
|  | SLINN+PH+RC(1M) | 69 | 4.96 | 1.37 | 70 | 0.025 | 25.9 |
|  | SLINN+PH+RC(DM) | 67.3 | 5.12 | 1.40 | 73 | 0.026 | 23.6 |
|  | LAAKSO(DM) | 71.8 | 1.23 | 0.53 | 91 | 0.006 | 5.4 |
| Coarse mode | SLINN | 1184 | 5.64 | 4.19 | 53 | 0.006 | 1.72 |
|  | SLINN+PH | 1182 | 5.26 | 4 | 54 | 0.005 | 1.6 |
|  | SLINN+PH+RC | 1194 | 5.25 | 4.01 | 54 | 0.005 | 1.58 |
|  | WANG | 1195 | 5.76 | 4.28 | 53 | 0.006 | 1.74 |
|  | LAAKSO | 1231 | 3.6 | 3.43 | 57 | 0.003 | 1.05 |
|  | SLINN+PH+RC(1M) | 1190 | 13.2 | 6.55 | 23 | 0.01 | 4 |
|  | SLINN+PH+RC(DM) | 1190 | 3.81 | 3.5 | 49 | 0.003 | 1.15 |
|  | LAAKSO(DM) | 1219 | 2.93 | 3.08 | 55 | 0.002 | 0.86 |

**Table 2. Global dust metrics split by mode (accumulation / coarse) for all of the simulations performed in this study**






A similar pattern emerges for the coarse mode, with ~30 % less coarse dust burden in LAAKSO than in SLINN+PH+RC (3.6 Tg compared to 5.25 Tg), owing to the greater scavenging rates for $1 \leq d_p \leq 2$ µm in LAAKSO (Fig. 3). The total dust burden is 53 % less in LAAKSO compared to SLINN+PH+RC (4.7 compared to 9.9 Tg) while DOD is 70 % smaller in LAAKSO than in SLINN+PH+RC (0.009 compared to 0.029). For perspective, the SLINN+PH+RC global-mean total DOD of 0.029 is

commensurate to the Aerocom phase 1 mean DOD of 0.029, the intermodel mean DOD from Kok *et al.* (2021) of 0.028, the mean DOD from most of the CRESCENDO models (Checa-Garcia *et al.*, 2021), and the observationally constrained range of 0.02-0.035 from Ridley *et al.* (2016). The total dust lifetime is 2.8 days in SLINN+PH+RC, 1.3 days in LAAKSO, and 4 days in WANG, which can be compared to a multimodel mean of 2.5 (+/- 1.3) days in the CRESCENDO models (Checa-Garcia *et al.*, 2021). This tentatively suggests that the SLINN+PH+RC dust metrics are closest to other state-of-the art climate models

and observations, whilst LAAKSO may underestimate the longevity of dust in the atmosphere and WANG may overestimate it. However, a range of caveats limits the extent to which we can say one BCS model is superior to another (see Section 6).

Figure 7 shows a comprehensive range of dust metrics for the SLINN+PH+RC, WANG, and LAAKSO simulations including spatial maps of annual mean dust burden (Figs 7a-c); seasonal DODs against observations from Kok *et al.* (2021) (circles) and

AERONET sites (pluses) (Figs 7d-f); seasonal near-surface dust concentrations against U-MIAMI observations (Figs 7g-i); and annual dust deposition against observations compiled by Huneeus *et al.* (2011) (Figs 7j-l). The scatter plots (Figs 7d-l) are supplemented by 3 statistical measures of predictive skill: the mean correlation coefficient (*r*), the mean bias, and the root mean square error (RMSE), all of which are calculated in logarithmic (base 10) space owing to the measurements ranging over many orders of magnitude. These statistics are meant to complement the figures and illustrate the closeness of fit between the

model and observations, but do not necessarily show which BCS model is best owing to competing biases and other caveats (see Section 6). Spatial plots of annual-mean values for each of the four observation data sets are shown in Fig. S6 in the Supplement. It is clear from all of the observational datasets in Fig. S6 that dust is prevalent over source regions in North and Equatorial Africa, the Middle East, and Asia, and less prevalent over the Americas, South Africa, much of the Pacific Ocean, and the Poles.


Dust is widely distributed over the Earth in WANG, with the greatest burden in the Northern Hemisphere (NH) but substantial concentrations in the Southern Hemisphere (SH) (Fig. 7b). Conversely, dust is almost entirely confined to the NH in LAAKSO, with only source regions in South Africa, South America, and Australia (Fig. S7 in the Supplement) exhibiting substantial dust burdens in the SH (Fig. 7c). Dust burdens in SLINN+PH+RC are intermediate between LAAKSO and WANG. Simulated

DOD in LAAKSO is vastly less than both AERONET and Kok *et al.* (2021) observations, particularly over secondary source regions such as South America, South Africa, and Australia. Furthermore, dust concentrations away from source regions (Fig. 7i) and deposition rates over the Pacific Ocean (Fig. 7l) are significantly less in LAAKSO than in the observations, which may imply that the LAAKSO BCS scheme is removing dust too efficiently from the atmosphere near to source regions. Conversely, WANG appears to overestimate dust away from source regions (Fig. 7h), despite all models exhibiting too little dust over



**Figure 7. Global dust metrics in the SLINN+PH+RC, WANG, and LAAKSO simulations, used to answer KQ1 – empirical vs theoretical BCS schemes. (a-c) annual-mean total dust burden, (d-f) seasonal and regional dust optical depths (DOD) against 440n nm AERONET observations (+) and 550nm DOD from Kok *et al*. (2021), (g-i) seasonal and regional near surface dust concentrations against U-MIAMI observations (Prospero and Nees, 1986), (j-l) annual-mean regional dust deposition rates against observations from Huneeus *et al*. (2011)**







source regions such as the Sahara, which is reflected in uniformly negative DOD biases relative to AERONET (Figs. 7d-f). Underestimating Saharan dust emissions (or at least, DOD) appears to be a persistent problem in Met Office Hadley Centre climate models (Mulcahy *et al.*, 2018), and will be exacerbated here by the fact that the largest and smallest dust bins in the 605 existing dust scheme (CLASSIC) are not resolved in UKCA dust.

The WANG simulation exhibits the smallest bias and RMSE in all the metrics (Fig. 7). However, this is partly due to positive biases away from source regions offsetting negative biases closer to dust source regions. The SLINN+PH+RC simulation exhibits a good spread about the 1:1 line in terms of comparing simulated DOD and dust concentrations with observations, 610 albeit with a slight overall negative bias which may emanate from insufficient dust emissions (Figs 7d,g). However, dust deposition rates over the Southern Ocean (SOc in Fig. 7j) are somewhat overestimated in SLINN+PH+RC which may emanate from spuriously elevated dust emissions in Australia and Southern Africa as also exhibited by UKESM (Checa-Garcia *et al.*, 2021), although note that the dust emissions scheme differ somewhat between UKESM and UM-GA8.0 (Woodward *et al.*, 2022). Given the many facets of the dust scheme which may contribute to dust distribution biases, such as deficiencies in 615 emissions and dry deposition rates and precipitation biases, it is impossible to pronounce value judgement on which BCS scheme is best from these simulations. However, it is possible to conclude that dust spatial distributions are highly sensitive to the choice of BCS scheme, with LAAKSO removing dust much more efficiently than SLINN+PH+RC or WANG, and closer to source regions (Fig. S8 in the Supplement).

**5.4 KQ2: Importance of missing processes in the Slinn (1984) BCS model**

Figure 8 shows the same dust metrics as in Fig. 7 but plotted for the UM-GA8.0 simulations based on the Slinn (1984) BCS scheme, with and without the missing processes of phoresis and rear capture. The global dust burden is significantly greater in SLINN (27.6 mg m$^{-2}$) than in SLINN+PH (19.5 mg m$^{-2}$) or SLINN+PH+RC (19.3 mg m$^{-2}$), which is mostly driven by an enhanced accumulation mode dust burden (Table 1). As accumulation mode aerosol is more optically active at the 440 nm spectral wavelength measured by AERONET than the coarse mode, this results in a reduced DOD bias in SLINN (-0.06, Fig. 625 8d) compared to SLINN+PH (-0.49, Fig. 8e), or SLINN+PH+RC (-0.51, Fig. 8f). However, dust concentrations away from source regions are greater in SLINN than in the observations (Fig. 8g), and the positive bias in dust deposition rate over the Southern Ocean is also exacerbated in SLINN, suggesting that the BCS rates may be too low in that model (Fig. 8j). Other confounding factors affect the atmospheric transport of the dust, such as dry deposition, particle shape, and in-cloud scavenging, and so it is impossible to definitely say that the BCS rates in SLINN are wrong, only that dust is removed less 630 efficiently by BCS in that model, which logically follows from the differences in scavenging rates (Fig. 3).



**BCS sensitivity to missing processes in Slinn (1984)**

**Figure 8.** The same as Fig. 7 but for SLINN, SLINN+PH, and SLINN+PH+RC simulations and used to answer KQ2 – impact of missing processes in the Slinn (1984) BCS algorithm





From Fig. 8, it is clear that phoresis has a significant impact on simulated dust concentrations, particularly in the removal of accumulation mode aerosol. The addition of rear-capture to the model has a smaller impact in the UM-GA8.0 simulations than the addition of phoresis. However, GCMs are unable to resolve heavy precipitation episodes owing to their coarse spatiotemporal resolution (Frei *et al.*, 2006), and are beset with annual and seasonal precipitation biases. For instance, the previous generation Met Office Hadley Centre climate model (UM-GA7.0) exhibited negative annual-mean precipitation

biases over the Indian subcontinent and in general overestimated precipitation over the oceans (Walters *et al.*, 2019), with many of the precipitation issues unrectified in UM-GA8.0 (Figure S9 in the Supplement). Given that the rear-capture effect increases in magnitude non-linearly with precipitation intensity (Fig. 3), it is likely rear-capture plays a more important role in wet removal of accumulation mode dust than exhibited in these simulations; a hypothesis which could be tested using a higher resolution climate model or a Numerical Weather Prediction (NWP) model. Additionally, precipitation biases will feed through to the dust metrics in Fig. 8, which again reduces our ability to bestow value judgement on the various SLINN schemes other

than to rank them in terms of dust removal rates. It is clear from Fig. 8 that models using the Slinn (1984) BCS scheme without consideration for phoresis and to a lesser extent rear-capture may be significantly underestimating wet removal of aerosol.

### 5.5 KQ3: Single-moment vs double moment BCS schemes

A double moment BCS scheme, wherein separate scavenging coefficients are applied to the zeroth (number) and third (mass)

moments of the aerosol size distribution accounting for the width of the aerosol mode, will differ most from a single moment BCS scheme wherever the number and mass scavenging coefficients differ most from the uniform scavenging coefficient (Fig. 4). All of the UM-GA8.0 simulations apart from SLINN+PH+RC(1M) employ a double-moment BCS approach for mineral dust (Table 1). SLINN+PH+RC(1M) instead uses the *Slinn+ph+rc* BCS model as in SLINN+PH+RC but applies uniform scavenging coefficients ($\Lambda$) to both number and mass concentrations rather than number ($\Lambda_N$) and mass ($\Lambda_M$) scavenging

coefficients separately. Because of this, the mineral dust size distributions are not permitted to evolve following BCS in SLINN+PH+RC(1M), which is the same approach used in the default UKCA BCS scheme (applied to all aerosols).

From Fig. 4, it is expected that the wider coarse mode ($\sigma = 2$) would be more affected by the double moment approach compared to the single moment approach than the narrower accumulation mode ($\sigma = 1.59$), which proves to be the case in the

UM-GA8.0 simulations. Nevertheless, the accumulation dust burden is 7.5 % greater in SLINN+PH+RC(1M) than in SLINN+PH+RC (Table 1), and the lifetime of the dust aerosol is 2 days greater in SLINN+PH+RC(1M) than in SLINN+PH+RC (26 compared to 24 days). Thus, the impact of using a double moment approach on accumulation mode aerosol should not be discounted. The impact on the coarse mode is more pronounced, with the dust lifetime increasing from 1.6 days in SLINN+PH+RC to 4 days in SLINN+PH+RC(1M) resulting in a factor of 2.5 increase to coarse mode dust burden

in SLINN+PH+RC(1M) (Table 2).





Figure 9 shows the same global and seasonal dust metrics for the SLINN+PH+RC, SLINN+PH+RC(1M), and SLINN+PH+RC(DM) simulations as in Figs 7 and 8. Interestingly many of the statistical measures of skill relative to the observations are better in the SLINN+PH+RC(1M) simulation than in SLINN+PH+RC, for example, for surface concentrations the RMSE is significantly less in SLINN+PH+RC(1M) (6.17 compared to 10.65 in SLINN+PH+RC) and negative DOD biases are also reduced. This is rather surprising, given that the double moment scheme is more physically plausible than the simple single moment approach, and again highlights the sensitivity of aerosol schemes in GCMs to many interwoven processes such as size distribution assumptions, emissions, sedimentation, and underlying meteorological biases. Like WANG (Fig. 7) and SLINN (Fig. 8), SLINN+PH+RC(1M) exhibits too much dust deposition over the Southern Ocean (Fig. 9k) which may be attributable to positive dust emission biases in regions such as Australia, South America, and South Africa such as seen in UKESM1, although note that the dust emission schemes are not precisely the same between UKESM1 and UM-GA8.0 (Checa-Garcia *et al*., 2021) and to inefficient wet removal rates in SLINN+PH+RC(1M). Over dust source regions such as the Sahara, negative biases in DOD (Fig. 9d) and surface concentrations (Fig. 9g) in SLINN+PH+RC are significantly reduced in SLINN+PH+RC(1M) (Figs 9e and 9h respectively), but this again may be an artefact of competing model biases, namely inefficient wet removal of dust and inaccurate dust emissions or representative size distribution. For instance, Mulcahy *et al*. (2018) found a low DOD bias over the Sahara in simulations with UM-GA8.0 and UKESM1. Thus, a qualified answer to Key Question 3 is that the double moment approach does have a significant impact on simulated dust concentrations compared to a single moment approach, in particular enhancing wet removal rates of the wide coarse mode aerosol.

## 5.6 KQ4: Impacts of representing downward mode-merging

The downward mode merging scheme applied in SLINN+PH+RC(DM) and LAAKSO(DM) redistributes aerosol mass and number from the coarse insoluble mode to the accumulation insoluble mode when the coarse mode median diameter falls below a fixed diameter threshold, in this case $d_p = 1$ μm (Mann *et al*., 2010). Recall that mode-merging is used to artificially represent the contraction of the coarse mode due to size dependent loss processes such as BCS (i.e., Fig. 6) or sedimentation which is difficult in models with fixed modal widths. In particular, the "sedimentation-driven" downward merging will happen nearer sources and one result of it will be that the size distribution of the dust which reaches the area where BCS occurs will be changed from the control experiments. The total mass transferred from the coarse to the accumulation mode is 10.8 Tg yr$^{-1}$ in SLINN+PH+RC(DM) and 9.8 Tg yr$^{-1}$ in LAAKSO(DM), or 0.9 and 0.8 % of the primary coarse dust emissions respectively. Most of the mode merging takes place near to sources regions over the Sahara, Middle East, and East Asian deserts and within 3-5 km of the surface (Fig. S10 in the Supplement). The coarse dust lifetime is reduced from 1.58 days in SLINN+PH+RC to 1.15 days in SLINN+PH+RC(DM), and 1.05 days in LAAKSO to 0.86 days in LAAKSO(DM). Concomitantly, the coarse dust burden decreases by 27 % in SLINN+PH+RC(DM) compared to SLINN+PH+RC, and 19 % in LAAKSO(DM) compared to LAAKSO, with corresponding increases in accumulation burden in the downward mode-merging simulations (Table 1).








**Figure 9. The same as Figs 7-8, but for SLINN+PH+RC, SLINN+PH+RC(1M), and SLINN+PH+RC(DM). Used to answer KQ3 – single vs double moment BCS schemes, and KQ4 – impact of representing downward mode-merging**







**Figure 10. Dust volume size distributions in a cross Atlantic transect in the SLINN+PH+RC and SLINN+PH+RC(DM) simulations for (a) June conditions in the region (58-61 ºW, 11-14 ºN) and 2-2.4 km altitude compared to SALTRACE measurements, (b) August conditions in the region (18-24 ºW, 14-24 ºN) and 2-3 km altitude compared to AER-D measurements, and (c) June conditions in the region (4-8 ºW, 21-26 ºN) and 0.1-1.2 km altitude compared to Fennec 2011 measurements. (d) shows the horizontal boundaries of the averaging regions in the Equatorial Atlantic**

Clearly mode-merging has a sizeable impact on the distribution of dust mass between the 2 modes. Figure 9 shows the spatial dust metrics in the SLINN+PH+RC and SLINN+PH+RC(DM) simulations and Fig. S11 in the Supplement shows the equivalent metrics for the LAAKSO and LAAKSO(DM) simulations. It is clear that downward mode-merging has a negligible impact on the overall dust metrics, for instance, the spatial distribution and magnitude of the total dust burden are similar in SLINN+PH+RC (Fig. S9a) and SLINN+PH+RC(DM) (Fig. S9c). The statistical measures of model fit compared to the observations, such as the biases and RMSEs, are equally comparable between the two simulations.


The main difference between the simulations becomes apparent when plotting the particle size distributions (PSDs). Figure 10 shows the volume size distributions (VSDs) for an equatorial cross-Atlantic transect, with simulated PSDs from SLINN+PH+RC and SLINN+PH+RC(DM) directly compared to observations from 3 independent summer (June-August) field campaigns. As a caveat, it is not possible to quantify how representative the observations are of the regional- mean dust PSDs, given that the aircraft campaigns measure a small sample space both spatially and temporally, but often remain our only datasets to measure the vertical structure of regional atmospheric aerosol. Figure S12 in the supplement shows the equivalent

VSDs for LAAKSO and LAAKSO(DM), and Fig. S13 in the supplement shows the number size distributions (NSDs) for all four simulations and observations.

It is clear from Fig. 10 that a significant amount of dust volume over the Saharan source region is missing in both SLINN+PH+RC and SLINN+PH+RC(DM) (Fig. 10c), which is at least partially caused by the inability of the current UKCA-

mode scheme to represent super-coarse dust emissions. In order to rectify this, a third insoluble mode representing super-coarse dust aerosol may in future be added to UKCA-mode. Simulated VSDs for the accumulation and coarse modes ($d_p <$ 10 μm) are in good agreement with Fennec 2011 observations over the Sahara (Fig. S10c). Over the east Atlantic, the median diameter of the coarse mode is significantly greater in SLINN+PH+RC(DM) than in SLINN+PH+RC, which agrees better with the AER-D VSD observations (Fig. 10b), albeit with a large difference in absolute coarse mode VSD which is likely

linked to the lack of super-coarse dust emissions (Fig. 10c). Finally, over the west Atlantic, the SALTRACE observations indicate a significant quantity of coarse mode dust advected from the Sahara, which is not apparent in either simulation and may again be related to the inability to represent super-coarse dust emissions (Fig. 10a). Nevertheless, the median diameter of the coarse mode is in better agreement with SALTRACE in SLINN+PH+RC(DM) than in SLINN+PH+RC and considerable coarse mode mass is preserved in SLINN+PH+RC(DM) (Fig. 10a). In summary, Fig. 10 shows that downward mode merging

acts to preserve coarse mode mass during atmospheric transport and effectively counteract the lack of contractability of modes, which is an artefact of the double-moment modal architecture. Therefore, in answer to KQ4, it may be important to represent downward mode merging in modal aerosol schemes that resolve particle growth and contraction processes such as BCS, in order to correctly resolve the aerosol PSDs.

## 6 Conclusions and Discussion

In this paper, various widely used parameterisations of the below cloud scavenging (BCS) of aerosol by rain droplets are presented and directly compared in climate simulations with the Met Office's Unified Model (UM-GA8.0). In particular, a new parameterisation is presented for the collection efficiency of particles due to rear capture in the wake of falling rain droplets, which can be added to the established collection efficiencies due to Brownian motion, inertial impaction, interception, thermophoresis, diffusiophoresis, and electric charge effects (Wang *et al.*, 2010). It is found that rear-capture is the dominant





BCS loss process for accumulation size particles under moderate to heavy rainfall conditions but has less of a cumulative impact on simulated dust concentrations in UM-GA8.0 than the addition of the three phoretic processes alone.

Four outstanding key questions (KQs) pertinent to numerical BCS schemes are answered in this paper. Namely: what is the impact of using empirical rather than theoretical BCS schemes? (KQ1); how important are missing processes to the ubiquitous
Slinn (1984) BCS scheme? (KQ2); what is the impact of using a single-moment rather than double-moment BCS approach? (KQ3); and how important is it to represent mode-merging alongside BCS in modal aerosol models? (KQ4). To answer these KQs, 20-year simulations using UM-GA8.0 were performed where the only variable is the underlying BCS scheme applied to UKCA-mode mineral dust aerosol. BCS scavenging coefficients were calculated offline and tabulated for simple interpolation as function of aerosol median diameter, modal width, and ambient rain rate online in UKCA-mode. UKCA-mode mineral dust
aerosol was selected because of its high potential for improvement, given that simulated dust concentrations are persistently too high in the default UKCA-mode dust setup which has often been attributed to inefficient wet deposition processes. It is therefore an ideal aerosol candidate for this type of sensitivity study.

Our simulations have highlighted the high sensitivity of simulated dust aerosol to the choice of BCS scheme, for example,
accumulation mode dust lifetime ranged from 5.4 days (LAAKSO) to 43.8 days (SLINN) and coarse mode dust lifetime ranged from 0.9 days (LAAKSO(DM)) to 4 days (SLINN+PH+RC(1M)). In answer to KQ1, the use of empirically derived BCS rates significantly underestimates dust concentrations and deposition rates away from source regions compared to observations (LAAKSO), whilst the theoretical BCS model exhibited dust concentrations comparable with observations (SLINN+PH+RC) (Fig. 7). This tentatively corroborates Wang *et al.* (2010, 2014)'s suggestion that the best BCS model to use in GCMs is the
theoretical model with the greatest scavenging rates (i.e., *Slinn+ph+rc*). Interestingly, Wang *et al.* (2014)'s semi-empirical model exhibits dust concentrations that are too high away from source regions in these simulations (WANG, Fig. 7). The statistical measures of fit (in particular, the bias and RMSE) used to compare simulated and observed DOD, surface dust concentrations, and deposition rates, suggest that the WANG simulation may be closer to observations than LAAKSO or SLINN+PH+RC overall, but this appears to be a result of competing errors, i.e., too little dust near source regions and too
much dust away from source regions. Given that the *Wang* scheme was fit to theoretical BCS models before the parameterisation of the rear-capture effect existed and given that Wang *et al.* (2014) implicitly used many of the *Slinn+ph+rc* parameterisations in their formulation of the *Wang* model, the use of the more physical *Slinn+ph+rc* BCS scheme in aerosol models appears to be the most accurate approach of those tested here.

In answer to KQ2, the addition of phoresis to the Slinn (1984) BCS model has a significant impact on simulated accumulation mode dust burden akin to a halving globally (SLINN+PH vs SLINN, Table 1 and Fig. 8). The addition of rear-capture on top of phoresis to SLINN has a more muted impact than phoresis alone which may be underestimated here given the inability of coarse resolution GCMs to resolve heavy precipitation episodes, and the non-linear increase in rear-capture collection





efficiency with rain rate (Fig. 3). Additionally, we have only tested the impacts of representing rear-capture alongside phoresis
(SLINN+PH) and not on its own (i.e., SLINN+RC). If we had run SLINN+RC, it is probable that rear-capture would have a
significant impact on the dust concentrations compared to SLINN, given the dominance of rear-capture over a large swathe of
the aerosol and raindrop size distributions (Fig. 2). In summary, neglecting the processes of rear-capture and phoresis in the
*Slinn* model may significantly overestimate submicron-sized (i.e., accumulation mode) dust burdens.

KQ3 and KQ4 are particularly pertinent to modal aerosol schemes, which are widely employed by GCMs (Gliß *et al.*, 2021).
In answer to KQ3, the use of a single-moment BCS approach (applied to a double moment aerosol scheme) which does not
account for modal width has a small impact on the narrow accumulation mode but a large impact on the broad coarse mode.
For example, the global coarse dust burden increases from 5.3 Tg in SLINN+PH+RC to 13.2 Tg in SLINN+PH+RC(1M)
(Table 1). Therefore, a single moment BCS scheme (as employed by default in UKCA-mode) may significantly underestimate
the wet deposition of coarse mode aerosol. In answer to KQ4, downward mode merging has little overall impact on total dust
concentrations in this model (SLINN+PH+RC(DM) vs SLINN+PH+RC, Fig. 9), but does have a significant impact on the
partitioning of dust between the accumulation and coarse insoluble modes (Fig. 10). Given the structural limitation in the
double moment modal aerosol approach, i.e., the fixed mode width, downward mode-merging may be a useful method to
reconcile simulated and observed aerosol size distributions.


Although the primary aim of this study is to impartially compare various BCS schemes from the literature in an appropriate
GCM framework with all else being equal, the stimulation for such a study was the inadequate performance of the existing
UKCA-mode dust scheme compared to observations and to the default UM-GA8.0 dust scheme (CLASSIC). An interesting
Supplementary Question (SQ) is then: How do the global dust metrics compare between a simulation with the new double-
moment BCS setup (using the *Slinn+ph+rc* BCS model) and simulations with CLASSIC and with the default UKCA-mode
dust scheme? To provide a preliminary answer to this question, which will be answered in more detail in a follow-on paper,
the same configuration of UM-GA8.0 was employed as in the rest of this study for one simulation with CLASSIC dust in its
default setup (6 bins) (Woodward *et al.*, 2022), and one for UKCA-mode dust with its existing single-moment BCS scheme
(Mann *et al.*, 2010). Global dust metrics in the SLINN+PH+RC simulation are compared to CLASSIC and default UKCA in
Fig. 11. Although it is unclear whether the single-moment BCS approach is culpable for the inferior performance in UKCA
(default) away from dust source regions, given the many facets of the dust scheme, it is clear that simulated dust surface
concentrations are markedly closer to observations away from source regions in SLINN+PH+RC than in UKCA (default) and
are now comparable with CLASSIC dust. Although our tests have focused on AMIP simulations in a climate configuration,
the efficiency of the new *Slinn+ph+rc* BCS scheme and the improved dust performance (Fig. 11) now makes UKCA-mode
dust a candidate for global NWP simulations with the UM (e.g., Mulcahy *et al.*, 2014).





## SLINN+PH+RC vs CLASSIC and UKCA (default)

**Figure 11.** The same as Fig. 7 but used to compare the SLINN+PH+RC simulation with UM-GA8.0 with CLASSIC dust (left column) and UM-GA8.0 with UKCA-mode dust and its default BCS scheme. This is used to answer the Supplementary Question: the impact of the *Slinn+ph+rc* BCS scheme versus existing UM-GA8.0 schemes





This work has focused on BCS models for aerosol schemes. While we have shown that including a more theoretically based BCS model significantly improves the simulation of dust (e.g., comparing SLINN+PH+RC with LAAKSO, Fig. 7), we are not arguing that this is a panacea for dust modelling. For example, the work presented here uses just 2 modes to represent atmospheric dust and this is not sufficient to resolve the observed size distribution near source regions, which leads to significant underestimation of volume associated with missing super-coarse particles and number associated with small Aitken particles (e.g., Fig. 10). This may be partially rectified in future by the addition of a third insoluble mode to represent super-coarse dust. Secondly, the ageing of dust from interaction with soluble atmospheric aerosols is not represented in the simulations, and therefore dust is not able to act as liquid CCN here (i.e., in-cloud scavenging), which is potentially an important atmospheric sink for mineral dust (Rodríguez *et al.*, 2021). Even its purely insoluble state, dust may act as CCN according to Köhler theory, which is not accounted for in these simulations. In this work the ageing scheme is not switched on but in the future work will be undertaken to assess the role of ageing in UKCA-mode dust simulations. Despite their limitations, UM-GA8.0 and UKCA-mode remain state-of-the-art climate and chemistry/aerosol models respectively (Sellar *et al.*, 2019) and are ideally placed as a framework to perform such as an investigation as documented here.

The BCS scheme developed here has only been tested with one aerosol type (mineral dust), and in future it would be informative to test the scheme with other aerosols (e.g., sulphate, black carbon, organic carbon, sea-salt). In particular, soluble aerosol may be less sensitive to the underlying BCS model given its ability to act as CCN and therefore be efficiently removed from the atmosphere via ICS (Haywood and Boucher, 2000). The results also may differ if a model with a higher spatiotemporal resolution is employed given the non-linear propensity of aerosol 'rear-capture' to rain rate (Fig. 3) and the ability of a high-resolution model to resolve heavy precipitation episodes. Additionally, the BCS models described in Section 2 were processed offline assuming standard atmospheric conditions and making assumptions on, e.g., the relationship between cloud droplet number density and rainfall rate, with the results tabulated and then used for simple interpolation in UKCA-mode. This is a computationally efficient method of evaluating BCS but does not account for differences in temperature, pressure, humidity, raindrop electric charge or other atmospheric variables which all affect BCS rates. Jung *et al*. (2003), Berthet *et al*. (2010), and Croft *et al*. (2010) offer numerical methods to explicitly evaluate BCS rates online which may be a more refined and exact if computationally expensive approach. Finally, we've explored BCS for aerosol capture by liquid rain droplets, but the current BCS scheme in UKCA-mode for aerosol capture by snow crystals is also a simple single-moment approach (Mann *et al*., 2010). Given the large differences between dust in the single and double moment BCS schemes (e.g., Fig. 9), it will be instructive to also improve the BCS scheme for snow, which may have a substantial impact on dust concentrations at high latitudes and in mountainous regions.

This study provides a summary of numerical modelling approaches for the below-cloud scavenging of aerosol by liquid rain droplets and answers key questions concerning the implications of selecting one BCS scheme over another. It is found that the simulated accumulation mode dust lifetime ranges from 5.4 days using an empirical BCS scheme (LAAKSO) to 43.8 days

using a theoretical scheme (SLINN) while the coarse mode dust lifetime ranges from 0.9 days (LAAKSO(DM)) to 4 days (SLINN+PH+RC(1M)), which highlights the high sensitivity of dust concentrations to BCS scheme. Given the wide range of BCS rates from the different empirical and theoretical models, it would be useful to the aerosol modelling community to further constrain the range of BCS rates using laboratory experiments, and to determine whether the disparity between the observed
and theoretical BCS rates is truly due to confounding atmospheric processes.

## Code Availability

Due to intellectual property rights restrictions, we cannot provide either the source code or documentation papers for the UM. The Met Office Unified Model is available for use under licence. A number of research organisations and national meteorological services use the UM in collaboration with the Met Office to undertake basic atmospheric process research,
produce forecasts, develop the UM code, and build and evaluate Earth system models. For further information on how to apply for a licence, see http://www.metoffice.gov.uk/research/modelling-systems/unified-model (last access: 4 May 2022). The *Slinn+ph+rc* BCS scheme is now available on the 'trunk' (the Met Office's data repository) and is available for all future UM versions since vn12.2.

## Data Availability

UM output used to produce Table 2 and Figures 7-10 is available from the Centre of Environmental Data Analysis (CEDA) at http://dx.doi.org/10.5285/2e36fe8eb7ee4bd0a0833d3e1edd795a (Jones *et al.*, 2022). Python and Fortran scripts used to produce the figures and tables of BCS rates are available from Zenodo at https://doi.org/10.5281/zenodo.6617052 (Jones, 2022).

## Competing Interests

The authors declare that they have no conflict of interest.

## Author contributions

ACJ developed the BCS model with assistance from AH, JH, PL, and AQ. JH assisted in implementing the scheme in UM-GA8.0. PL and AQ developed a new formula for the collision efficiency due to rear-capture from their own laboratory data. SW and CR assisted in analysing the results of the simulations including the provision of plotting scripts. ACJ wrote the
manuscript with assistance from all co-authors.



**Acknowledgements**

The authors would like to thank the Met Office and the UM team for providing the UM climate model; Joseph Prospero and the University of Miami for freely providing dust concentration data; all the principal investigators of AERONET for their free provision of aerosol retrieval data; and Bernadett Weinzierl for freely providing SALTRACE dust PSDs. Figures were produced using Python 3.6.10 (https://www.python.org/) and Iris 2.4.0 (https://scitools.org.uk/).

This work and its contributors (ACJ and AH) were supported by the UK-China Research & Innovation Partnership Fund through the Met Office Climate Science for Service Partnership (CSSP) China as part of the Newton Fund. SW was supported by the Met Office Hadley Centre Climate Programme funded by BEIS.

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
