# Peer review of "Below-cloud scavenging of aerosol by rain: A review of numerical modelling approaches and sensitivity simulations with mineral dust"

_Atmospheric Chemistry and Physics, 2022_

## Referee Comment (RC1)

**Reviewer's report on the manuscript by Jones et al. "Below-cloud scavenging of aerosol by rain: A review of numerical modelling approaches and sensitivity simulations with mineral dust", Atmospheric Chemistry and Physics, Manuscript ID: acp-2022-409**

The manuscript presents an examination of exiting parameterizations for below-cloud scavenging (BCS) of aerosols by rain in the context for use in GCMs, particularly pertaining to those with a modal representation of aerosols. Simulations of mineral dust using a GCM (the UK Met Office's Unified Model coupled with UKCA-mode for chemistry and aerosols) were conducted, employing a number of theoretical and empirical formulations of BCS rate. The study aims at addressing several questions: 1) the impact of using an empirical vs. a theoretical formulation for the BCS rate on GCM modelled (mineral dust) aerosols, given the large difference in BCS rates between the two approaches; 2) the importance of the additional physical processes that are often missing in existing BCS parameterizations, such as phoresis and rear-capture processes; 3) the impact of assuming monodispersed aerosols in calculating the BCS rate rather than integrated BCS rates to account for the lognormal distribution of modal aerosols; and 4) the impact of mode merging following BCS. The last two questions are relevant to models using a modal representation of aerosol size distribution only. In addition, the authors also proposed a new parameterization for collection efficiency to account for the rear-capture mechanism. The large uncertainty in the parameterization of BCS of aerosols by hydrometeors remains to be resolved. This study explores the sensitivity of GCM simulated dust aerosols to the different formulations for the BCS rates for size-distributed aerosols. The manuscript is well structured. I do however have some concerns and comments (see below) which I hope that the authors can address before the manuscript can be published.

General comments

The sensitivity results from the GCM simulations of mineral dust can be influenced by how some of the other processes are represented in the model. For example, the choice of treating mineral dust particles as externally mixed insoluble particles throughout their atmospheric lifetime in this study limits the wet removal of dust particles to BCS only. However, through atmospheric processing, dust particles can be coated and become internally mixed with other soluble components. They can participate in cloud process and be subjected to in-cloud scavenging (ICS or rainout). The ICS can be particularly important for accumulation mode aerosols at greater distance downwind from the source regions. If atmospheric aging and ICS were considered for mineral dust in the model, would the sensitivity to BCS be reduced?

How is BCS modelled in the UKCA-mode for other soluble and insoluble modes? Would the same Slinn+ph+rc BCS algorithm be used for those aerosol modes also?

A question on the comparison of modelled and observed (measured) dust deposition fluxes (in Figures 7, 8, and 9): are those total deposition fluxes (model and observation), i.e., including both dry (including sedimentation) and wet deposition fluxes? If so, what is the dry-vs-wet dust deposition fluxes based on the model simulations?

Specific comments

Line 110: KQ4 is not just relevant for BCS. With a modal representation for aerosol size distribution, mode merging will need to be considered for any process that is size dependant. Does the default UKCA-mode not consider model merging for such processes as dry deposition and sedimentation, cloud processing, coagulation, in addition to wet deposition?

Lines 388-389: Is mode merging not performed by default in UKCA-mode (following those processes that affect aerosol size distributions)?

Line 451: Figure 2 is intended to illustrate the relative importance of various physical processes/mechanisms contributing to the overall collection efficiency and BCS rate over the range of particle and rain droplet sizes. However, how do you define dominance here? Is it the one with largest collection efficiency numerically (since it only identifies a single process at any given particle and droplet size)? It would be more instructive to show the contributions from the various processes/mechanisms to the total collection efficiency over the particle size spectrum at some given droplet size (e.g., 0.1 and 1 mm, perhaps). How does the result here compare with the review of Wang et al. (2010) (i.e., their Fig. 1); is the colour label switched over between interception and inertial impaction? Also, the area where the rear-capture mechanism is dominant on Figure 2 seems to be pasted on; the contours of the total collection efficiency seem to be discontinuous there.

Lines 471-472: Note that the 90$^{th}$ percentile fit of Wang et al. (2014) also accounted for the variability from droplet number density and fall velocity formulations.

Lines 491-493: The discussion on aerosol median diameter converging over time is unclear. The authors seem to be referring to the crossover between $\Lambda_N$ and $\Lambda_M$ shown on Figure 4.

Lines 505-506: It is curious that Figure 5(b) shows the least change in the accumulation mode diameter after 3-hour integration for Slinn+ph+rc and Slinn+ph+rc(1M), while Figure 5(a) shows the most mass loss for Slinn+ph+rc and Slinn+ph+rc(1M) amongst the non-observation-based BCS schemes. Any explanations?

Lines 590-591: The authors made a comment here about the simulated DOD from LAAKSO being significantly biased low compared with observations, particularly over secondary source regions. The low bias is apparent from Figure 7(f); however, the specific locations of the low bias is not obvious from the said figure.

Lines 592-594: Again the authors are referring to Fig. 7(h) and (i) for the discussion here on where the modelled surface dust concentrations are biased low or high from LAAKSO and WANG compared to observations. However, such information is not indicated from these figures (unlike the scatter plots for dust deposition).

Lines 625-626: Same here, how can you tell the from Fig. 8(g) that the modelled dust concentrations (SLINN in this case) are higher than observations away from source regions?

Lines 658-660: It is not just the wider model width for the coarse mode that are attributable to the greater difference in model results between the double moment vs. single moment approach. The accumulation mode covers the range of particle spectrum where the overall BCS rates are less sensitive to particle size than the size range where the coarse mode covers (ref. to Figure 3).

Line 756: Again, KQ4 is not just relevant to BCS.

Lines 759-761: How is BCS modelled in the default UKCA-mode dust setup?

Lines 775-777: Wang et al. (2014) did not include any rear-capture parameterization in developing their semi-empirical model.

Lines 781-784: The relatively muted effect of the rear-capture mechanism (with regard to the modelled dust metrics) may also be consistent with the relatively narrow range in the droplet sizes when the mechanism is important as shown in Figure 2, as well as possible buffering effects of the multiple processes in the model influencing the overall simulation results.

Lines 785-787: Suggest removing this statement as the argument here is not a reasonable one. The overall collection efficiency is a linear combination of all the collection efficiencies representing each individual physical processes/mechanism in BCS. The inclusion of the phoresis processes would not mask the contribution from the rear-capture mechanism.

Lines 787-788: Table 2 seems to indicate that the significant reduction in modelled global accumulation-model dust burden is mainly due to the addition of the phoresis processes, (rather than a combination of phoresis and rear-capture).

Line 805: What is the BCS scheme used in CLASSIC?

Lines 810-813: How can you tell the model over-predictions of the dust surface concentration in the scatterplot (Figure 11h) are over areas away from the dust source regions?

Line 859-860: Do we know whether the observational derived BCS rates are free of ICS influence?

Minor comments

It would be good to be consistent in referring to $E_{br,in,im,th,df,es,rc}$ as collection efficiencies throughout the text rather than switching between collection and collision efficiencies at different places.

Equation 3a: should it be $N(D_d) = N_0 e^{-\lambda D_d}$?

Line 484: Replace "uniform size distribution" with "monodispersed aerosols"?

Line 485: You mean Eqs 4 and 5 (not 3 and 4)?

Lines 487-488: Could be reworded to "It is clear that the *effective* number ($\Lambda_N$) and mass ($\Lambda_M$) scavenging coefficients for lognormal aerosol distribution are significantly greater than the scavenging coefficient ($\Lambda$) for monodispersed aerosols, …".

Line 732: Referring to Fig. S12c rather than Fig. S10c?

Line 774: Use "compensating errors" rather than "competing errors"?

---

## Referee Comment (RC2)

**Review ACP journal**

Below-cloud scavenging of aerosol by rain: A review of numerical modelling approaches and sensitivity simulations with mineral dust

article reference: acp-2022-409

**1 Overview**

**1.1 Topic and relevance**

The manuscript shows a set of sensitivity studies of below-cloud scavenging (BSC) modelling schemes for the case of mineral dust. Given the current challenges to converge both: empirical studies and modelling approaches, and the potential impact of improvements in the global climate models, the presented study is relevant enough to be published in Atmospheric Chemistry and Physics (ACP) journal.

**1.2 Evaluation**

The manuscript is well written, with an useful set of results, and providing a review with a substantial amount of information about both, BCS models and the Met Office's Unified Model (UM) set up. I recommend to **accept the paper with minor revisions**. Below the authors will find few suggestions organised by sections of the paper after two minor general comments.

**1.3 General comments**

- Although somehow the information is already included in **Table 2**, for the mineral dust community might be useful to see role of each BSC scheme in the estimations of wet deposition fluxes in $[\text{Tg yr}^{-1}]$. For example, in (Shao et al., 2011) and (R. Checa-Garcia et al., 2021) the estimations of wet deposition fluxes in $[\text{Tg yr}^{-1}]$ show large differences between models, but also the ratio Dry/Wet deposition.

- Note that for mineral dust emissions using nudged surface wind speeds from reanalysis increase the consistency between models (vs winds fields from each model). Here is not much relevant because it is a single model analysis but, anyway, it is worth to mention which wind fields are the authors considering as dust burden and dust emission can be conditioned by this fact.

**2 Comments/Suggestions**

**2.1 Title**

The title is descriptive but, given that the sensitivity studies rely on Unified Model, I wonder if the authors would consider to add this information in the title (it's not a requirement, just an idea)

**2.2 Abstract**

- **Line 21**: Maybe a ',' before while

- **Line 25-26**: This fact is not covered by the title.

**2.3 Introduction**

- **Line 85**: Given that the paper is a kind of review paper, with a wider spectrum of possible readers, I would double check about consistency in nomenclature. Sometime it is used BCS rates, sometimes $\Lambda$, others BSC coefficient.

- **Line 130**: Maybe the authors can comment already here about single vs double moment approach in UM in consistency with KQ3 and KQ4. Is the *mode merging* in the UM introduced inside the BSC scheme? In other words, is it separated from other processes like sedimentation, for example? I understand that the idea is to compare simulation with and without *mode merging* but only for BSC (not other removal mechanisms).

**2.4 Section 2**

Here when authors describe the equations, remember that mineral dust are not usually spherical, so $d_p$ represents (probably) an effective diameter. For large raindrops, the shape is also not spherical (as you commented in the paper there are oscillations), so also $D_p$ would represent an effective diameter.

- **Line 152**: BSC coefficient vs BSC rate?

- **Line 158**: Here I would add something like: involving two reasonable approximations one for diameters $D_p >> d_p$ and one for falling velocities $v_t(D_p) >> v_t(d_p)$.

- **Line 160**: Note that here you write $E(d_p, D_p)$ and in other sections $E(D_p, d_p)$.

- **Equation 3.a**: Probably is $N_0 e^{-\lambda D_d}$ ($\lambda$ not k).

- **Line 174**: rain droplet vs raindrops

- **Line 174**: The (Abel et al., 2012) new raindrop size distribution (DSD) seems to have more impact in low rainfall rate surface precipitation rather than larger rainfall rate. So, are the differences between (Abel et al., 2012) and previous DSD parameterization an important factor for the BSC rates? Or this only means a second order factor of discrepancies? In other words: it is likely that few models are using older parametrizations, even using the Marshall-Palmer (1948) model. Would be this an important aspect?

- **Note Section 2.2**: I would refer now (in the main paper) to the Table S2 in the supplement.

- **Note Section 2.3**: Here I would mention that later on you refer as **phoresis** the join role of all these three collection efficiencies; or the section title can be **Phoresis: ...** to be clear.

- **Equation 12**: Here it might be more readable to include the two cases like:

$$E_{rc}(d_p, D_p) = \begin{cases} ... & 20 \le R_{e,D} \le 800 \\ 0 & \text{otherwise} \end{cases}$$

**2.5 Section 3**

- **Line 315**: Arbitrary -> idealised?

- **Line 317**: I would add in the caption of the figures that it was a box-model simulation.

**2.6 Section 4**

- **Line 341**: For me it is unclear if both aerosols and chemistry are evaluated once per model hour or only chemistry (later on you mention about time-steps of 30 min/15 min for deposition)

- **Table 1**: As commented before, please explain that **phoresis** means all the processes described in section 2.3.

**2.7 Section 5**

- **Figure 2**: I understood that this figure shows the X, for which $E_X(d_p, D_p)$ has the highest values (either because X is more important or because the other mechanism decrease). Is this right? or **dominant** means significant larger contribution than other processes?

- **Line 462**: If here the authors are using the box-model simulations it would be worth to remember.

- **Line 484/Figure 4**: So I understood $\Lambda$ from (eq.2) and $\Lambda_{N,M}$ for (eq. 4 and eq.5) for a distribution with **geometric** median diameter and given $\sigma$?

- **Figure 3**: The parameterization of Wang shown in Figure 3 seems to me consistent with the figure 3 of the original paper of Wang (2014), as they considered one semi-empirical model with **phoresis**, i.e. from (Andronache et al., 2006), (but not rear-capture) it seems that in their estimations **phoresis** is only relevant in higher rainfall rates. Regarding the **Laasko model**, the results seems to me reasonable as somehow they have a linear dependence with rainfall rate in their parameterization (eq. 14b).

- **Figure 7**: This is a very interesting figure with a lot of information. However, if I understood well the observations of dust optical depths (DOD) of AERONET are mostly located at Sahel region and represented by + , and the DOD of (**Kok2021**) are regional averages over close-to-sources regions represented by circles. Is that right? Then (**Kok2021**) are not observations but a **constrained multi-model by observations**. In the figure it is not so easy to see the +. It would be great to have open circles for Southern hemisphere in (d), (e) and (f). This will help the reader to understand better the results.

- **Figure 8**: This figure is also interesting, as expected the inclusion of **phoresis** and **rear-capture** decrease the optical depths and surface concentrations. It is more difficult to detect differences in dust deposition, which might be because we have to signals here: wet and dry deposition. It would be interesting to compare with specific measurements of wet deposition flux (for example, Marticorena et al., 2017).

- **Figure 11**: Here like in Figure 7, it would be good to use open circles for Southern hemisphere.

**2.8 Supplementary information**

- **Note 1**: consider **raindrop** instead of **rain droplet**, in the precipitation community is used raindrop-size distribution rather than droplet (it is common cloud droplet but not cloud drop).

- **Note 2**: note that Eq.S23 is correct but often are considered two cutoffs and the integral is expressed by:

$$R = C \int_{D_{min}}^{D^{max}} D^3 N(D) v(D) dD$$

this is because drops smaller than a $D_{min}$ are not falling, and drops larger than $D_{max}$ are eventually broken into smaller drops (or not even formed) and this can have a effect in the rainfall properties, for example (Checa-Garcia et al., 2014).

**References**

Andronache, C. et al. (2006). "Scavenging of ultrafine particles by rainfall at a boreal site: observations and model estimations". In: *Atmospheric Chemistry and Physics* 6.12, pp. 4739–4754. DOI: 10.5194/acp-6-4739-2006. URL: https://acp.copernicus.org/articles/6/4739/2006/.

Shao, Yaping et al. (2011). "Dust cycle: An emerging core theme in Earth system science". In: *Aeolian Research* 2.4, pp. 181–204. ISSN: 1875-9637. DOI: https://doi.org/10.1016/j.aeolia.2011.02.001. URL: https://www.sciencedirect.com/science/article/pii/S1875963711000085.

Abel, S. J. and I. A. Boutle (2012). "An improved representation of the raindrop size distribution for single-moment microphysics schemes". In: *Quarterly Journal of the Royal Meteorological Society* 138.669, pp. 2151–2162. DOI: https://doi.org/10.1002/qj.1949. eprint: https://rmets.onlinelibrary.wiley.com/doi/pdf/10.1002/qj.1949. URL: https://rmets.onlinelibrary.wiley.com/doi/abs/10.1002/qj.1949.

Checa-Garcia, R et al. (2014). "Binning effects on in-situ raindrop size distribution measurements". en. In: *Atmos. Meas. Tech. Discuss.* 7.3, pp. 2339–2379. DOI: doi:10.5194/amtd-7-2339-2014.

Marticorena, B. et al. (2017). "Mineral dust over west and central Sahel: Seasonal patterns of dry and wet deposition fluxes from a pluriannual sampling (2006–2012)". In: *Journal of Geophysical Research: Atmospheres* 122.2, pp. 1338–1364. DOI: https://doi.org/10.1002/2016JD025995. URL: https://agupubs.onlinelibrary.wiley.com/doi/abs/10.1002/2016JD025995.

Checa-Garcia, R. et al. (2021). "Evaluation of natural aerosols in CRESCENDO Earth system models (ESMs): mineral dust". In: *Atmospheric Chemistry and Physics* 21.13, pp. 10295–10335. DOI: 10.5194/acp-21-10295-2021. URL: https://acp.copernicus.org/articles/21/10295/2021/.

---

## Author Comment (AC1)

**Response to reviewers for paper: "Below-cloud scavenging of aerosol by rain: A review of numerical modelling approaches and sensitivity simulations with mineral dust" by A C Jones et al.**

We thank the Reviewers for their very useful comments and suggestions. We are glad that the Reviewers see the merits of the study and we have endeavoured to address all of the suggestions they have made below. Below, we have listed each of the Reviewer's comments in red, our replies in black, and highlighted changes to the manuscript in blue. All line references refer to the updated manuscript.

**Reviewer 1**

**General Comments**

➢ The sensitivity results from the GCM simulations of mineral dust can be influenced by how some of the other processes are represented in the model. For example, the choice of treating mineral dust particles as externally mixed insoluble particles throughout their atmospheric lifetime in this study limits the wet removal of dust particles to BCS only. However, through atmospheric processing, dust particles can be coated and become internally mixed with other soluble components. They can participate in cloud process and be subjected to in-cloud scavenging (ICS or rainout). The ICS can be particularly important for accumulation mode aerosols at greater distance downwind from the source regions. If atmospheric aging and ICS were considered for mineral dust in the model, would the sensitivity to BCS be reduced?

This is an important caveat which we have already discussed in Section 6 (L870): "Secondly, the ageing of dust from interaction with soluble atmospheric aerosols is not represented in the simulations, and therefore dust is not able to act as liquid CCN here (i.e., in-cloud scavenging), which is potentially an important atmospheric sink for mineral dust (Rodríguez *et al*., 2021). Even its purely insoluble state, dust may act as CCN according to Köhler theory, which is not accounted for in these simulations." Further to this sentence, we have added the following to L873:

Therefore, the sensitivity of dust deposition to the choice of BCS scheme may be overestimated in these simulations given that ICS processes are not accounted for.

➢ How is BCS modelled in the UKCA-mode for other soluble and insoluble modes? Would the same Slinn+ph+rc BCS algorithm be used for those aerosol modes also?

These questions have already been answered in Section 6 (L880): "The BCS scheme developed here has only been tested with one aerosol type (mineral dust), and in future it would be informative to test the scheme with other aerosols (e.g., sulphate, black carbon, organic carbon, sea-salt). In particular, soluble aerosol may be less sensitive to the underlying BCS model given its ability to act as CCN and therefore be efficiently removed from the atmosphere via ICS (Haywood and Boucher, 2000)."

➢ A question on the comparison of modelled and observed (measured) dust deposition fluxes (in Figures 7, 8, and 9): are those total deposition fluxes (model and observation), i.e., including both dry (including sedimentation) and wet deposition fluxes? If so, what is the dry-vs-wet dust deposition fluxes based on the model simulations?

Both the modelled and observed dust deposition fluxes are "total" fluxes (i.e., wet+dry). The following clarification has been added to L433.

A global network of dust total deposition fluxes (i.e., involving wet and dry deposition processes) is provided by Huneeus *et al.* (2011).

Furthermore, we show the global-mean wet deposition fractions for all of the simulations in Table 2 and discuss these values in Section 5.3 (L578), e.g., "It is clear that BCS is significantly greater in LAAKSO, with 89 % of accumulation dust removed by wet deposition compared to only 72 % in SLINN+PH+RC and 52 % in WANG.".

**Specific Comments**

➢ Line 110: KQ4 is not just relevant for BCS. With a modal representation for aerosol size distribution, mode merging will need to be considered for any process that is size dependant. Does the default UKCA-mode not consider model merging for such processes as dry deposition and sedimentation, cloud processing, coagulation, in addition to wet deposition?

➢ Lines 388-389: Is mode merging not performed by default in UKCA-mode (following those processes that affect aerosol size distributions)?

The default UKCA-mode aerosol setup includes upwards mode merging for the soluble modes following aerosol growth processes such as cloud processing, coagulation, and condensation, but does not include downward mode merging. The following clarification has been added to the manuscript (L394):

The default UKCA-mode aerosol setup includes upward mode merging for the soluble modes following aerosol growth processes such as cloud processing, coagulation, and condensation, but does not represent downward mode merging. Note that only a subset of simulations described here include downward mode-merging (see Table 1).

➢ Line 451: Figure 2 is intended to illustrate the relative importance of various physical processes/mechanisms contributing to the overall collection efficiency and BCS rate over the range of particle and rain droplet sizes. However, how do you define dominance here? Is it the one with largest collection efficiency numerically (since it only identifies a single process at any given particle and droplet size)? It would be more instructive to show the contributions from the various processes/mechanisms to the total collection efficiency over the particle size spectrum at some given droplet size (e.g., 0.1 and 1 mm, perhaps). How does the result here compare with the review of Wang et al. (2010) (i.e., their Fig. 1); is the colour label switched over between interception and inertial impaction? Also, the area where the rear-capture mechanism is dominant on Figure 2 seems to be pasted on; the contours of the total collection efficiency seem to be discontinuous there.

We define dominance as the largest collection efficiency determined using the algorithm described in Section 2 and standard atmospheric conditions. The following has been added to the caption for Fig. 2.

(i.e., the largest determined numerically)

The following has been added to Section 5.1 (L458):

… where by 'dominant' we mean the largest collection efficiency numerically determined using the algorithm described in Section 2 and standard atmospheric conditions (Table S1 in the Supplement)

Figure 2 serves to show the importance of the rear-capture effect (e.g., we say in L455 "Given that a new formulation for the 'rear-capture' collection efficiency is provided in this paper (Eq. 12), it is also useful to assess if and when rear-capture makes an important contribution to the overall collection efficiency"), which we think it does effectively. Other papers have shown the contributions to the overall collection efficiency at certain droplet sizes rather than over the spectrum of droplet sizes, such as Wang *et al.* (2010), their Fig. 1, as the Reviewer has noted. We do not feel it is necessary to repeat this exercise, nor to match the colour scheme in Wang *et al.*'s Fig. 1, as this paper is not a direct follow on to Wang *et al*. The rear-capture effect is not pasted on – the contours of the total collection efficiency are discontinuous because this is the upper droplet diameter for the legitimacy of the formula for $E_{rc}(D_d, d_p)$ (i.e., Eq. 12). The following has been added to Section 5.1 (L464):

Note that the contours of the total collection efficiency are discontinuous at $D_d \approx 2 \times 10^3$ µm in Fig. 2 because this is the upper droplet diameter for the legitimacy of the formula for $E_{rc}(D_d, d_p)$ (i.e., Eq. 12), above which $E_{rc} = 0$.

➤ Lines 471-472: Note that the 90th percentile fit of Wang et al. (2014) also accounted for the variability from droplet number density and fall velocity formulations.

We thank the reviewer for the suggestion. The following has been added to L484:

… although the *Wang* model also accounted for the variability from droplet number density and fall velocity formulations.

➤ Lines 491-493: The discussion on aerosol median diameter converging over time is unclear. The authors seem to be referring to the crossover between $\Lambda_N$ and $\Lambda_M$ shown on Figure 4.

This is not a trivial process and so some further discussion is warranted and has been added to the text at L508:

The change in aerosol median diameter over a timestep can be related to the scavenging rates $\Lambda_N$ and $\Lambda_M$ using Eq. 15 (where $\overline{d_{p,0}}$ is the median diameter at the start of the timestep and $\Delta t$ is the timestep in seconds).

$$\Delta \overline{d_p} = \overline{d_{p,0}} \left( e^{\frac{(\Lambda_N - \Lambda_M)}{3} \Delta t} - 1 \right) \begin{cases} \Delta d_p < 0 & \Lambda_N < \Lambda_M \\ \Delta d_p = 0 & \Lambda_N = \Lambda_M \\ \Delta d_p > 0 & \Lambda_N > \Lambda_M \end{cases} \qquad (\text{Eq. } 15)$$

Therefore, although both the number and mass concentration decrease each timestep, the median diameter may increase, decrease, or remain the same depending on the scavenging rates $\Lambda_N$ and $\Lambda_M$ and will ultimately converge to a value of $d_p$ such that $\Lambda_N = \Lambda_M$.

➤ Lines 505-506: It is curious that Figure 5(b) shows the least change in the accumulation mode diameter after 3-hour integration for Slinn+ph+rc and Slinn+ph+rc(1M), while Figure 5(a) shows the most mass loss for Slinn+ph+rc and Slinn+ph+rc(1M) amongst the non-observation-based BCS schemes. Any explanations?

This relates to the convergence diameter (i.e., $d_p$ such that $\Lambda_N = \Lambda_M$) being different for the different BCS schemes. Clearly the convergence diameter is closer to the initial conditions (i.e., $\overline{d_p} = 0.4$ µm)

for the Slinn+ph+rc model than for the other models, but at the same time $\Lambda_M$ and $\Lambda_N$ are greater for Slinn+ph+rc than for the other theoretical BCS schemes.

> ➢ Lines 590-591: The authors made a comment here about the simulated DOD from LAAKSO being significantly biased low compared with observations, particularly over secondary source regions. The low bias is apparent from Figure 7(f); however, the specific locations of the low bias is not obvious from the said figure.

This is very good point, and is also relevant for Figures 8, 9, 11, and S11 in the Supplement and their associated analysis. To overcome this issue, we have decided to group the observations by approximate region, including by hemisphere. Tables S3-S5 in the Supplement (i.e., the observations) give which region each observation is ascribed too. Rather than differentiate the observations by season in Figs 7, 8, 9, 11 and S11, we now differentiate by region (as we already did with Figs j-l), as the differentiation by season was not actually referred to in the text and was thus superfluous. Long names for the abbreviated regions are provided in the Supplement, with only the abbreviations given in the figures and when referred to in the text. An example figure (Fig. 7) is below.

[Figure]

We now explicitly refer to these regions in the text throughout the Results section, e.g., in L618:

"secondary source regions such as South America, South Africa, and Australia (SAm, SAf, and Aus respectively in Figs 7d-f)."

We thank the Reviewer for the suggestion as this change has significantly improved the manuscript.

> Lines 592-594: Again the authors are referring to Fig. 7(h) and (i) for the discussion here on where the modelled surface dust concentrations are biased low or high from LAAKSO and WANG compared to observations. However, such information is not indicated from these figures (unlike the scatter plots for dust deposition).

> Lines 625-626: Same here, how can you tell the from Fig. 8(g) that the modelled dust concentrations (SLINN in this case) are higher than observations away from source regions?

Please see the answer above

> Lines 658-660: It is not just the wider model width for the coarse mode that are attributable to the greater difference in model results between the double moment vs. single moment approach. The accumulation mode covers the range of particle spectrum where the overall BCS rates are less sensitive to particle size than the size range where the coarse mode covers (ref. to Figure 3).

We thank the reviewer for the suggestion. The following has been added to L694:

owing to the greater mode width and the fact that the accumulation mode covers the range of particle spectrum where the overall BCS rates are less sensitive to particle size

> Line 756: Again, KQ4 is not just relevant to BCS.

The following caveat has been added to L794:

Note that while mode-merging is investigated here in the context of BCS, it may be equally applicable to other atmospheric aerosol loss processes.

> Lines 759-761: How is BCS modelled in the default UKCA-mode dust setup?

The CLASSIC and UKCA (default) BCS schemes are not described in detail here as that is not the primary aim of the paper. The following sentence has been added to L844 to refer the reader to details of the schemes:

For full descriptions of the existing UKCA-mode and CLASSIC BCS schemes see Mann *et al.* (2010) and Woodward *et al.* (2001), respectfully.

> Lines 775-777: Wang *et al.* (2014) did not include any rear-capture parameterization in developing their semi-empirical model.

We already refer to this in the text (L814): "Given that the Wang scheme was fit to theoretical BCS models before the parameterisation of the rear-capture effect existed and given that Wang *et al.* (2014) implicitly used many of the *Slinn+ph* parameterisations in their formulation of the Wang model, …"

> ➢ Lines 781-784: The relatively muted effect of the rear-capture mechanism (with regard to the modelled dust metrics) may also be consistent with the relatively narrow range in the droplet sizes when the mechanism is important as shown in Figure 2, as well as possible buffering effects of the multiple processes in the model influencing the overall simulation results.

This is a good point; we thank the reviewer for the suggestion and have adopted the text in full. The following has been added to the manuscript at L823:

The relatively muted effect of the rear-capture mechanism (with regard to the modelled dust metrics) may also be consistent with the relatively narrow range in the droplet sizes when the mechanism is important as shown in Figure 2, as well as possible buffering effects of the multiple processes in the model influencing the overall simulation results.

> ➢ Lines 785-787: Suggest removing this statement as the argument here is not a reasonable one. The overall collection efficiency is a linear combination of all the collection efficiencies representing each individual physical processes/mechanism in BCS. The inclusion of the phoresis processes would not mask the contribution from the rear-capture mechanism.

This statement has been removed.

> ➢ Lines 787-788: Table 2 seems to indicate that the significant reduction in modelled global accumulation-model dust burden is mainly due to the addition of the phoresis processes, (rather than a combination of phoresis and rear-capture).

This is true and mentioned in the text in L819: "the addition of phoresis to the Slinn (1984) BCS model has a significant impact on simulated accumulation mode dust burden akin to a halving globally (SLINN+PH vs SLINN, Table 2 and Fig. 8). The addition of rear-capture on top of phoresis to SLINN has a more muted impact than phoresis"

> ➢ Line 805: What is the BCS scheme used in CLASSIC?

The CLASSIC and UKCA (default) BCS schemes are not described in detail here as that is not the primary aim of the paper. The following sentence has been added to L844 to refer the reader to details of the schemes:

For full descriptions of the existing UKCA-mode and CLASSIC BCS schemes see Mann *et al.* (2010) and Woodward *et al.* (2001), respectfully.

> ➢ Lines 810-813: How can you tell the model over-predictions of the dust surface concentration in the scatterplot (Figure 11h) are over areas away from the dust source regions?

As discussed earlier, we now group the observations by region and hemisphere in Figs 7, 8, 9, 11 and S11 which allows us to ascribe the dust surface concentration biases to regions. For this statement, we add the following reference (L857):

(e.g., over the Pacific Ocean, Pac, Fig. 11h)

> ➢ Line 859-860: Do we know whether the observational derived BCS rates are free of ICS influence?

This is an open question discussed by many papers (e.g., Wang *et al.*, 2010) as we mention in the Introduction (L60): "A recent hypothesis is that the enhanced BCS rates from observations may be due to contributions from ICS and other confounding atmospheric processes such as turbulent diffusion, given that it is difficult to conduct a controlled BCS experiment in the actual atmosphere (Andronache et al., 2006; Wang et al., 2011)."

**Minor comments**

> ➢ It would be good to be consistent in referring to Ebr,in,im,th,df,es,rc as collection efficiencies throughout the text rather than switching between collection and collision efficiencies at different places.

We thank the reviewer for the suggestion and have changed collision to collection throughout

> ➢ Equation 3a: should it be $N(Dd)=N0e^{-\lambda Dd}$?

We have made the change

> ➢ Line 484: Replace "uniform size distribution" with "monodispersed aerosols"?

We have made the change

> ➢ Line 485: You mean Eqs 4 and 5 (not 3 and 4)?

We have made the change

> ➢ Lines 487-488: Could be reworded to "It is clear that the effective number ($\Lambda N$) and mass ($\Lambda M$) scavenging coefficients for lognormal aerosol distribution are significantly greater than the scavenging coefficient ($\Lambda$) for monodispersed aerosols, …".

We thank the Reviewer for the suggestion and have adopted the text in its entirety (L500)

> ➢ Line 732: Referring to Fig. S12c rather than Fig. S10c?

We have made this change

> ➢ Line 774: Use "compensating errors" rather than "competing errors"?

We have made this change

**Reviewer 2**

**General Comments**

> ➢ Although somehow the information is already included in Table 2, for the mineral dust community might be useful to see role of each BSC scheme in the estimations of wet deposition fluxes in [Tg yr-1]. For example, in (Shao et al., 2011) and (R. Checa-Garcia et al., 2021) the estimations of wet deposition fluxes in [Tg yr-1] show large differences between models, but also the ratio Dry/Wet deposition.

As the Reviewer notes, the global-mean wet deposition fractions (%) for each simulation are provided in Table 2 and discussed briefly in Section 5.3. Additionally, spatial maps of the annual-mean total dust deposition rate are provided in Figure S8 in the Supplement, and speciated (wet/dry) monthly dust deposition data is provided for all of the simulations in the Supplementary dataset uploaded to CEDA: http://dx.doi.org/10.5285/2e36fe8eb7ee4bd0a0833d3e1edd795a.

> ➢ Note that for mineral dust emissions using nudged surface wind speeds from reanalysis increase the consistency between models (vs winds fields from each model). Here is not much relevant because it is a single model analysis but, anyway, it is worth to mention which wind fields are the authors considering as dust burden and dust emission can be conditioned by this fact.

The simulations are free running and not nudged to reanalyses. We have added the following to Section 4.3 (L407):

The simulations are free running (i.e., not nudged to reanalyses)

**Specific Comments**

> ➢ The title is descriptive but, given that the sensitivity studies rely on Unified Model, I wonder if the authors would consider to add this information in the title (it's not a requirement, just an idea)

We thank the Reviewer for the suggestion. The title is now:

Below-cloud scavenging of aerosol by rain: A review of numerical modelling approaches and sensitivity simulations with mineral dust in the Met Office's Unified Model

> ➢ Line 21: Maybe a ',' before while

We have made this change

> ➢ Line 25-26: This fact is not covered by the title.

Given the broad scope of this paper, it has not been possible to explicitly include all of its aims in the title. Instead, we think the umbrella term: "A review of numerical modelling approaches" encompasses the new parameterisation of the rear-capture effect, particularly since it is a revision of the parameterisation presented by Lemaitre *et al.* (2017)

> ➢ Line 85: Given that the paper is a kind of review paper, with a wider spectrum of possible readers, I would double check about consistency in nomenclature. Sometimes it is used BCS rates, sometimes Λ, others BSC coefficient.

This is a good point. We universally adopt the term 'BCS rate' (including in Figs 3-4) as it is more informative than 'scavenging coefficient'. Additionally, we omit the use of Λ in the Introduction.

> ➢ Line 130: Maybe the authors can comment already here about single vs double moment approach in UM in consistency with KQ3 and KQ4. Is the mode merging in the UM introduced inside the BSC scheme? In other words, is it separated from other processes like sedimentation, for example? I understand that the idea is to compare simulation with and without mode merging but only for BSC (not other removal mechanisms).

The following clarification has been added to the manuscript (L394):

The default UKCA-mode aerosol setup includes upward mode merging for the soluble modes following aerosol growth processes such as cloud processing, coagulation, and condensation, but does not represent downward mode merging. Note that only a subset of simulations described here include downward mode-merging (see Table 1).

> ➢ Here when authors describe the equations, remember that mineral dust are not usually spherical, so dp represents (probably) an effective diameter. For large raindrops, the shape is also not spherical (as you commented in the paper there are oscillations), so also Dp would represent an effective diameter.

This is a good point. We have added the following to L171:

Note that mineral dust particles are not usually spherical, so $d_p$ represents an effective diameter. For large raindrops, the shape is also not spherical so $D_d$ also represents an effective diameter.

> ➢ Line 152: BSC coefficient vs BSC rate?

BCS rate is now used throughout

> ➢ Line 158: Here I would add something like: involving two reasonable approximations one for diameters Dp >> dp and one for falling velocities vt(Dp) >> vt(dp).

We have added the following to L167:

The algorithm assumes two reasonable approximations, firstly that the diameter of the droplet is significantly greater than of the particle (D_d≫d_p) and secondly that the falling velocity of the droplet is significantly greater than for the particle (U_t (D_d )≫ U_t (d_p)).

> ➢ Line 160: Note that here you write E(dp; Dp) and in other sections E(Dp; dp).

We thank the Reviewer for identifying this inconsistency. We have changed all instances to dp; Dp throughout, including in the supplement

> Equation 3.a: Probably is $N_0e^{-\Lambda Dd}$ ($\Lambda$ not k)

We have made this change

> Line 174: rain droplet vs raindrops

We have changed all references to "rain droplets" to "raindrops" for consistency

> Line 174: The (Abel et al., 2012) new raindrop size distribution (DSD) seems to have more impact in low rainfall rate surface precipitation rather than larger rainfall rate. So, are the differences between (Abel et al., 2012) and previous DSD parameterization an important factor for the BSC rates? Or this only means a second order factor of discrepancies? In other words: it is likely that few models are using older parametrizations, even using the Marshall-Palmer (1948) model. Would be this an important aspect?

This is a good point. The following caveat has been added to the conclusions (L893):

The BCS scheme shown here employs a single parameterisation for the raindrop number density as a function of the rainfall rate from Abel and Boutle (2012) and a single parameterisation for the terminal velocity of falling droplets from Beard (1976). Wang et al. (2010) found that the choice of terminal velocity parameterisation could impact BCS rates by a factor of 2, and the choice of raindrop number density could impact BCS rates by a factor of 3-5. Therefore, the results presented here may be sensitive to the underlying parametrizations used for the raindrop properties

> Note Section 2.2: I would refer now (in the main paper) to the Table S2 in the supplement.

We already refer to Table **S1** in the Supplement (L218): "The overall formulae for the individual collection efficiencies are presented in Eqs 6-8 and the reader is referred to Section S3 and Table S1 in the Supplement for further details of the variables and their dependencies"

> Note Section 2.3: Here I would mention that later on you refer as phoresis the join role of all these three collection efficiencies; or the section title can be Phoresis: . . . to be clear.

We have made this change

> Equation 12: Here it might be more readable to include the two cases like:

$$E_{rc}(D_d, d_p) = \begin{cases} \dfrac{1}{1.37 \times 10^{10}}\, S_t^{-3.625} R_{e,D}^{1.444} e^{-0.243\,(\ln S_t)^2} e^{0.08144(\ln S_t)\ln R_{e,D}} & 20 \le R_{e,D} \le 800 \\ 0 & \text{otherwise} \end{cases} \quad (\text{Eq. } 12)$$

We thank the Reviewer for the suggestion. We have made this change

> Line 315: Arbitrary -> idealised?

We have made this change

➢ Line 317: I would add in the caption of the figures that it was a box-model simulation.

We have made this change

➢ Line 341: For me it is unclear if both aerosols and chemistry are evaluated once per model hour or only chemistry (later on you mention about time-steps of 30 min/15 min for deposition)

Both aerosols and chemistry are called once per model hour. This sentence has been updated (L347).

➢ Table 1: As commented before, please explain that phoresis means all the processes described in section 2.3.

We have made this change

➢ Figure 2: I understood that this figure shows the X, for which hEX(dp;Dp) has the highest values (either because X is more important or because the other mechanism decrease). Is this right? Or dominant means significant larger contribution than other processes?

We define dominance as the largest collection efficiency determined using the algorithm described in Section 2 and standard atmospheric conditions. The following has been added to the caption for Fig. 2.

(i.e., the largest determined numerically)

The following has been added to Section 5.1 (L458):

… where by 'dominant' we mean the largest collection efficiency numerically determined using the algorithm described in Section 2 and standard atmospheric conditions (Table S1 in the Supplement)

➢ Line 462: If here the authors are using the box-model simulations it would be worth to remember.

Figure 3 is not related to the box model simulations

➢ Line 484/Figure 4: So I understood $\Lambda$ from (eq.2) and $\Lambda N;M$ for (eq. 4 and eq.5) for a distribution with geometric median diameter and given ?

Yes. We have endeavoured to make this clearer through Section 2 and Section 3. Namely, we have added the dependencies of the BCS rates to the variables, e.g., $\Lambda(d_p, R)$, $\Lambda_N(\overline{d_p}, \sigma, R)$, and $\Lambda_M(\overline{d_p}, \sigma, R)$.

➢ Figure 3: The parameterization of Wang shown in Figure 3 seems to me consistent with the figure 3 of the original paper of Wang (2014), as they considered one semi-empirical model with phoresis, i.e. from (Andronache et al., 2006), (but not rear-capture) it seems that in their estimations phoresis is only relevant in higher rainfall rates. Regarding the Laakso model, the results seems to me reasonable as somehow they have a linear dependence with rainfall rate in their parameterization (eq. 14b)

The parameterisation of Wang in Fig. 3 is not inconsistent with Fig. 3 of Wang *et al.* (2014). For example, for $d_p = 0.1$ μm and $d_p = 1$ μm, and for rainfall rates of $R = 0.1$ mm/hr and $R = 1$ mm/hr, the scavenging coefficient is between $2 \times 10^{-7}$ and $6 \times 10^{-7}$ in Wang *et al.* (2014), and is ~$4 \times 10^{-7}$ in our Fig. 3 for the intermediate $R = 0.5$ mm/hr. We assume any differences between Wang and Slinn+ph in Fig. 3 emanate from the treatment of raindrop number density in Slinn+ph, wherein our scheme (Abel and Boutle, 2012) exhibits greater number of small droplets at smaller rain rates (e.g., Fig. S3 in the Supplement). We have added this caveat to the paper (L484):

although the Wang model also accounted for the variability from raindrop number density and fall velocity formulations

> Figure 7: This is a very interesting figure with a lot of information. However, if I understood well the observations of dust optical depths (DOD) of AERONET are mostly located at Sahel region and represented by + , and the DOD of (Kok2021) are regional averages over close-to-sources regions represented by circles. Is that right? Then (Kok2021) are not observations but a constrained multi-model by observations. In the figure it is not so easy to see the +. Itwould be great to have open circles for Southern hemisphere in (d), (e) and (f). This will help the reader to understand better the results

We thank the Reviewer for highlighting that Kok *et al.* (2021) data is actually observationally constrained simulation output, and we have changed references to Kok DODs to 'observationally constrained simulations' throughout the text. We have also grouped the observations by regions rather than season in Figs 7, 8, 9, 11, and S11 in the Supplement, which is reflected in the new color scheme. Additionally, we have taken the Reviewer's suggestion to make Southern Hemisphere measurements open circles rather than filled for Figs 7, 8, 9, 11 and S11 in the Supplement. We believe this change has significantly improved the manuscript.

> Figure 8: This figure is also interesting, as expected the inclusion of phoresis and rear-capture decrease the optical depths and surface concentrations. It is more difficult to detect differences in dust deposition, which might be because we have to signals here: wet and dry deposition. It would be interesting to compare with specific measurements ofwet deposition flux (for example, Marticorena et al., 2017).

This is an interesting proposition, and one we may return to in a follow up model development paper. At present, we continue to use the total dust deposition rates from Huneeus *et al.* (2011) which are widely used when evaluating dust models. As you say, the signal of reduced optical depths and surface concentrations is more pronounced between with and without phoresis, and thus supports our conclusion that phoresis has a significant impact on dust distribution.

> Figure 11: Here like in Figure 7, it would be good to use open circles for Southern hemisphere.

We have made this change

**Supplement**

> Note 1: consider raindrop instead of rain droplet, in the precipitation community is used raindrop-size distribution rather than droplet (it is common cloud droplet but not cloud drop).

We thank the Reviewer for the suggestion and have changed all instances of rain droplets to raindrops

> Note 2: note that Eq.S23 is correct but often are considered two cutoffs and the integral is expressed by:

$$R = C \int_{D_{min}}^{D_{max}} D_d{}^3 N(D_d) U_t(D_d) dD_d$$

this is because drops smaller than a Dmin are not falling, and drops larger than Dmax are eventually broken into smaller drops (or not even formed) and this can have a effect in the rainfall properties, for example (Checa-Garcia et al., 2014).

We thank the Reviewer for the information. The following has been added to the Supplement:

Note that often $R$ is expressed as the integral of $\frac{\rho_w \pi}{6} D_d{}^3 N(D_d) U_t(D_d) dD_d$ between threshold diameters (e.g., $D_{min}$ to $D_{max}$) to reflect the fact that very small raindrops do not drop, and very large raindrops eventually break into larger raindrops as they fall.